# Enrichment of *SARM1* alleles encoding variants with constitutively hyperactive NADase in patients with ALS and other motor nerve disorders

Jonathan Gilley[1]*, Oscar Jackson[1], Menelaos Pipis[2], Mehrdad A Estiar[3,4], Ammar Al-Chalabi[5,6], Matt C Danzi[7], Kristel R van Eijk[8], Stephen A Goutman[9], Matthew B Harms[10], Henry Houlden[2], Alfredo Iacoangeli[5,11,12], Julia Kaye[13], Leandro Lima[13,14], Queen Square Genomics[2], John Ravits[15], Guy A Rouleau[3,4,16], Rebecca Schüle[17], Jishu Xu[17], Stephan Züchner[7], Johnathan Cooper-Knock[18], Ziv Gan-Or[3,4,16], Mary M Reilly[2], Michael P Coleman[1]*

[1]John van Geest Centre for Brain Repair, Department of Clinical Neurosciences, University of Cambridge, Cambridge, United Kingdom; [2]Department of Neuromuscular Disease, UCL Queen Square Institute of Neurology and The National Hospital for Neurology, London, United Kingdom; [3]Department of Human Genetics, McGill University, Montreal, Canada; [4]The Neuro (Montreal Neurological Institute-Hospital), McGill University, Montreal, Canada; [5]Department of Basic and Clinical Neuroscience, Maurice Wohl Clinical Neuroscience Institute, Institute of Psychiatry, Psychology and Neuroscience, King's College London, London, United Kingdom; [6]Department of Neurology, King's College Hospital, King's College London, London, United Kingdom; [7]Dr. John T. Macdonald Foundation Department of Human Genetics and John P. Hussman Institute for Human Genomics, University of Miami Miller School of Medicine, Miami, United States; [8]Department of Neurology, UMC Utrecht Brain Center, University Medical Center Utrecht, Utrecht University, Utrecht, Netherlands; [9]Department of Neurology, University of Michigan, Ann Arbor, United States; [10]Institute for Genomic Medicine, Columbia University, New York, United States; [11]Department of Biostatistics and Health Informatics, Institute of Psychiatry, Psychology & Neuroscience, King's College London, London, United Kingdom; [12]National Institute for Health Research Biomedical Research Centre and Dementia Unit at South London and Maudsley NHS Foundation Trust and King's College London, London, United Kingdom; [13]Center for Systems and Therapeutics, Gladstone Institutes, San Francisco, United States; [14]Gladstone Institute of Data Science and Biotechnology, Gladstone Institutes, San Francisco, United States; [15]Department of Neurosciences, University of California, San Diego, La Jolla, United States; [16]Department of Neurology and Neurosurgery, McGill University, Montreal, Canada; [17]Center for Neurology and Hertie Institute für Clinical Brain Research, University of Tübingen, German Center for Neurodegenerative Diseases, Tübingen, Germany; [18]Sheffield Institute for Translational Neuroscience, University of Sheffield, Sheffield, United Kingdom

*For correspondence:
jg792@cam.ac.uk (JG);
mc469@cam.ac.uk (MPC)

**Abstract** SARM1, a protein with critical NADase activity, is a central executioner in a conserved programme of axon degeneration. We report seven rare missense or in-frame microdeletion human

*SARM1* variant alleles in patients with amyotrophic lateral sclerosis (ALS) or other motor nerve disorders that alter the SARM1 auto-inhibitory ARM domain and constitutively hyperactivate SARM1 NADase activity. The constitutive NADase activity of these seven variants is similar to that of SARM1 lacking the entire ARM domain and greatly exceeds the activity of wild-type SARM1, even in the presence of nicotinamide mononucleotide (NMN), its physiological activator. This rise in constitutive activity alone is enough to promote neuronal degeneration in response to otherwise non-harmful, mild stress. Importantly, these strong gain-of-function alleles are completely patient-specific in the cohorts studied and show a highly significant association with disease at the single gene level. These findings of disease-associated coding variants that alter SARM1 function build on previously reported genome-wide significant association with ALS for a neighbouring, more common *SARM1* intragenic single nucleotide polymorphism (SNP) to support a contributory role of SARM1 in these disorders. A broad phenotypic heterogeneity and variable age-of-onset of disease among patients with these alleles also raises intriguing questions about the pathogenic mechanism of hyperactive SARM1 variants.

## Editor's evaluation

This manuscript reports the discovery of an enrichment of SARM1 variants with constitutively active NADase activity among patients with ALS and related motor neuron diseases. The genetic evidence is substantiated by functional evidence showing consistent gain of function amongst the disease-associated variants. These findings draw renewed attention to a role for axonal degeneration in some forms of motor neuron disease and provide a rationale for SARM1-directed therapeutic intervention.

## Introduction

The toll-like receptor adaptor protein SARM1 is required for axon degeneration after injury, in response to toxins and other insults, and in several models of neurodegenerative disease (*Conforti et al., 2014*; *Coleman and Höke, 2020*). In many of these situations, the intimate interplay between pro-degenerative SARM1 (sterile alpha and TIR motif containing protein 1) (*Osterloh et al., 2012*; *Gerdts et al., 2013*) and its upstream, pro-survival regulator NMNAT2 (nicotinamide mononucleotide adenylyltransferase 2) (*Gilley and Coleman, 2010*; *Gilley et al., 2015*) is critical and most likely involves a disruption to normal nicotinamide adenine dinucleotide ($NAD^+$) homeostasis (*Gerdts et al., 2016*; *Figley and DiAntonio, 2020*) or related metabolites (*Angeletti et al., 2021*). NMNAT2 is one of three NMNAT isoforms that catalyse the last step in $NAD^+$ biosynthesis and is the predominant NMNAT in axons. The SARM1 TIR domain has a critical $NAD^+$ consuming multifunctional glycohydrolase (NADase) activity (*Essuman et al., 2017*; *Horsefield et al., 2019*; *Wan et al., 2019*) and, intriguingly, NMNAT substrate, NMN, and its product, $NAD^+$, can both influence this activity: NMN activates SARM1 NADase by binding to an allosteric site in the auto-inhibitory ARM domain of the enzyme (*Zhao et al., 2019*; *Figley et al., 2021*), whereas $NAD^+$ can oppose this activation by competing for binding to the same site (*Jiang et al., 2020*; *Sporny et al., 2020*; *Figley et al., 2021*). Rising NMN and declining $NAD^+$ as a consequence of loss of very short-lived NMNAT2 in damaged axons (*Di Stefano et al., 2015*) thus leads directly to activation of SARM1 NADase and a self-reinforcing $NAD^+$ decline that represents one possible cause of degeneration.

As well as its established role in axons, SARM1 can promote neuronal cell death under certain conditions. In most cases, the extent to which cell death and concurrent axon degeneration are primary or secondary events has not been fully elucidated (*Gerdts et al., 2013*; *Summers et al., 2014*; *Gerdts et al., 2015*; *Summers et al., 2016*; *Bratkowski et al., 2020*). However, recent studies do suggest direct involvement of SARM1 in neuronal cell death in some situations, including photoreceptor loss in models of retinal degeneration and Leber congenital amaurosis (LCA) (*Ozaki et al., 2020*; *Sasaki et al., 2020*) and as a result of Vacor neurotoxicity when specifically applied to cell bodies (*Loreto et al., 2021*).

Extensive evidence from animal models has implicated NMNAT-sensitive and SARM1-dependent axon degeneration and/or cell death in a variety of neurodegenerative disorders (*Conforti et al., 2014*; *Coleman and Höke, 2020*), but knowledge of genetic association with human diseases is more

limited. To date, biallelic *NMNAT2* loss-of-function (LoF) variants have been associated with two rare polyneuropathies that broadly resemble the corresponding mouse models (*Gilley et al., 2013*; *Gilley et al., 2019*; *Huppke et al., 2019*; *Lukacs et al., 2019*), and *NMNAT1* mutations cause photoreceptor loss in LCA via a mechanism involving SARM1 (*Sasaki et al., 2020*). While no firm association between *SARM1* variation and human disease has yet been established, genome wide association studies (GWAS) have linked the *SARM1* chromosomal locus, including an intragenic SNP, to sporadic ALS (*Fogh et al., 2014*; *van Rheenen et al., 2016*), although whether *SARM1* is the causative gene is not known.

Motor nerve disorders are neurodegenerative disorders that affect motor neurons in the brain, brainstem, and spinal cord. These include ALS, the most common adult-onset motor nerve disorder, as well as other inherited conditions like hereditary spastic paraplegia (HSP) (*Goutman, 2017*). There is also some phenotypic and genetic overlap with the hereditary motor neuropathies (*Rossor et al., 2012*). These disorders exhibit broad phenotypic heterogeneity and variable age-of-onset. Recently, SARM1-dependent death mechanisms have been found to play a significant role in a mutant TDP-43 model of ALS (*White et al., 2019*). This is important as TDP-43 aggregation is an almost universal pathologic feature in ALS (*Arai et al., 2006*; *Neumann et al., 2006*). While removal of SARM1 does not alleviate motor neuron degeneration in a *SOD1* model of ALS (*Peters et al., 2018*), this could be explained by the finding that TDP-43 aggregation is usually not seen in ALS linked to *SOD1* or *FUS* mutations (*Mackenzie et al., 2007*; *Maekawa et al., 2009*; *McAlary et al., 2019*). It therefore remains possible that SARM1 plays an important role in a majority of ALS cases.

Here, we have identified a number of rare alleles specific to ALS patients in Project MinE initiative data freeze 1 (*Project MinE ALS Sequencing Consortium, 2018*; *van der Spek et al., 2019*) and the Answer ALS project (*Rothstein et al., 2020*) encoding missense substitutions and in-frame microdeletions in the auto-inhibitory ARM domain of SARM1 that confer substantial pro-degenerative NADase gain-of-function (GoF). A number of these GoF alleles were subsequently identified in additional ALS patients and in HSP or other motor nerve disorder patients in other databases but, crucially, were not seen at all in any individuals in the control groups. Based on their strongly skewed distribution between patients and controls, we propose that these *SARM1* GoF alleles are thus strong candidates for risk alleles in ALS and other motor nerve disorders and raises the possibility that other mechanisms of SARM1 activation could make a wider contribution.

## Results

### Identification of a cluster of ALS patient-specific SARM1 ARM domain coding variants in Project MinE data freeze 1

Previous studies have shown that artificial missense mutations and in-frame deletions in the auto-inhibitory ARM domain of SARM1 are more likely to result in pro-degenerative GoF than disruption of its SAM multimerization domains or catalytic TIR domain, which instead is more likely to give rise to protective LoF (*Gerdts et al., 2013*; *Summers et al., 2016*; *Essuman et al., 2017*; *Horsefield et al., 2019*; *Jiang et al., 2020*; *Sporny et al., 2020*; *Bratkowski et al., 2020*; *Figley et al., 2021*; *Shen et al., 2021*). Therefore, if SARM1 contributes to ALS pathogenesis, we predicted that there might be a relative paucity of naturally occurring coding variation in the SAM and TIR domains and/or an enrichment of coding variation in the ARM domain among patients, with the opposite in unaffected individuals.

To explore this, we assessed the distribution of mostly rare *SARM1* missense SNP and small, in-frame deletion alleles between cases and controls in Project MinE consortium data freeze 1 (DF1), a public database containing WGS data for 4,366 mostly sporadic ALS patients and 1,832 non-ALS controls (*Project MinE ALS Sequencing Consortium, 2018*; *van der Spek et al., 2019*). Even accounting for the fact that a proportion of the coding variation is likely to be functionally neutral, the distribution of alleles was notably skewed (*Figure 1* and *Table 1*). Specifically, of the 16 alleles encoding changes between amino acids 112 and 385 of the SARM1 ARM domain, all are seen in patients, but only the two relatively more common alleles are also seen in controls, whereas the distribution of coding variation between patients and controls for the remainder of the gene is largely unremarkable, being much more proportional to the sizes of the patient and control groups. Based on this uneven distribution,

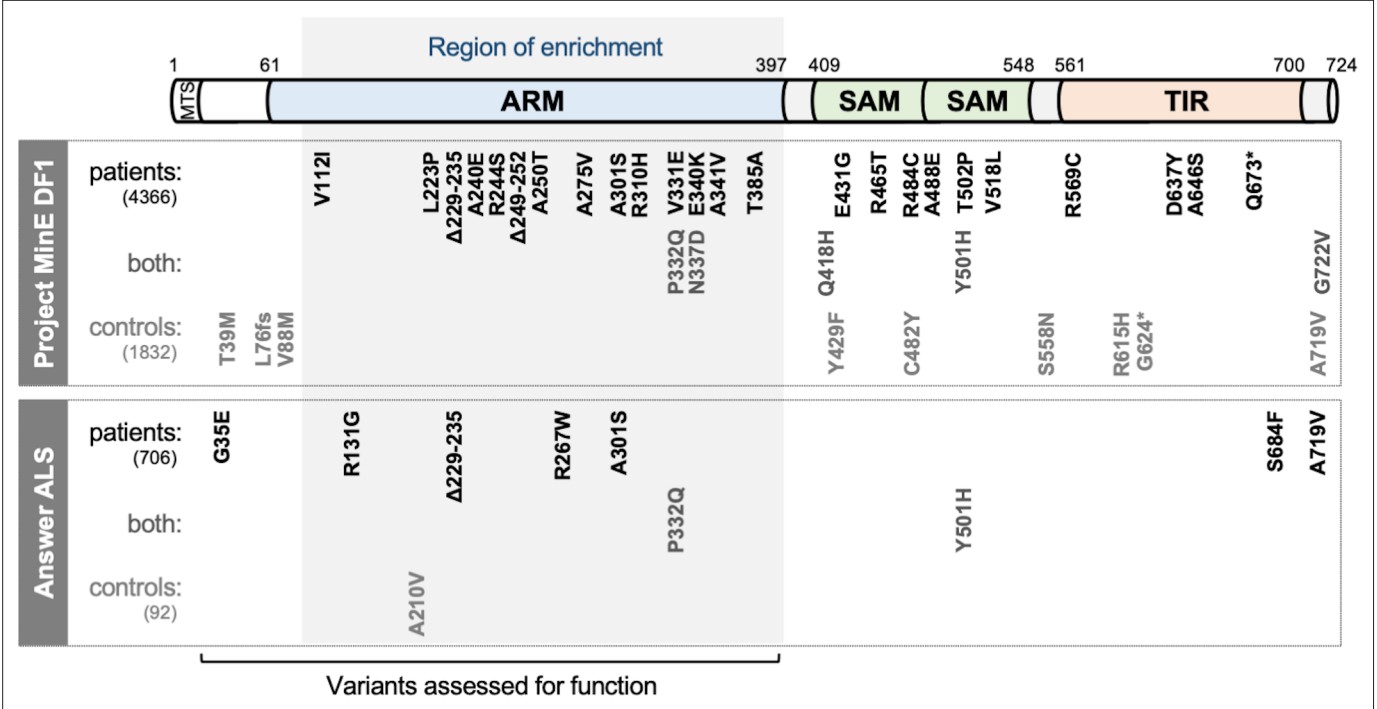

**Figure 1.** SARM1 coding variants in ALS patient and control groups. Schematic representation of the canonical 724 amino acid human SARM1 protein showing basic domain structure and the approximate locations of the coding variation present in the Project MinE DF1 and Answer ALS datasets. Variants are separated into those seen only in ALS patients, those seen in ALS patients and controls (both), and those seen only in controls. MTS, "mitochondrial" targeting sequence; ARM, HEAT/Armadillo motif domain; SAM, sterile alpha motif domains; TIR, Toll/Interleukin receptor domain. Special note: Variants are annotated according to the canonical 724 amino acid SARM1. Project MinE DF1 currently uses gnomAD v2.1.1, which is based on genome build GRCh37/ hg19, and annotates variants to a 690 amino acid product of a non-canonical reference *SARM1* cDNA sequence. This cDNA, and genome build GRCh37, contain single nucleotide substitutions, insertions and deletions at its 5' end when compared to gnomAD v3.1.1 and GRCh38/ hg38 which annotate to the canonical 724 amino acid SARM1. These differences mean that the 690 amino acid "isoform" has a truncated N-terminus, due to use of an alternative ATG codon, and switches frame until it eventually synchronises with 724 amino acid SARM1 at amino acid 107 (73 in the 690 amino acid '"isoform"). As such, numbering of SARM1 variants in Project MinE DF1 (and any databases annotated to gnomAD v2.11/ GRCh37) is 34 residues lower than that shown here over the synchronised region and differences more N-terminal to this must be determined individually by lift over from GRCh37 to GRCh38. Notably, the NCBI Single Nucleotide Polymorphism Database (dbSNP, including rsIDs) uses numbering based on the 724 amino acid SARM1 protein.

we thus hypothesised that some of the patient-specific Project MinE DF1 ARM domain alleles are risk alleles for ALS through pro-degenerative GoF.

## Several ALS patient-specific SARM1 ARM domain coding variants in Project MinE DF1 increase NAD⁺ depletion in transfected HEK 293T cells

We first tested the functional consequences of coding variation in the region spanning amino acids 112–385 of the SARM1 ARM domain by assessing the ability of variants to deplete $NAD^+$ in transfected HEK 293T cells. We hypothesised that some of them confer a harmful, and thus potentially pathogenic NADase GoF, but established assay conditions able to detect either GoF or LoF of the exogenously expressed SARM1 (*Figure 2—figure supplement 1A*). Importantly, any contribution from endogenous SARM1 in these assays was expected to be negligible due to its comparatively very low level of expression in the HEK 293T cell clone used (*Figure 2—figure supplement 1B*).

Under these conditions, $NAD^+$ levels are approximately halved in cells expressing exogenous WT SARM1 (due to overexpression of a low constitutive NADase activity) but we observed a substantially greater lowering of $NAD^+$ for six of the ALS-specific variants - L223P, Δ229–235 (alternatively named Δ226–232 - see *Table 1* legend), Δ249–252, V331E, E340K, and T385A SARM1 - suggestive of strong NADase GoF (*Figure 2A*). Smaller, but still statistically-significant lowering of $NAD^+$ was also seen for a further three variants - V112I, A275V and A341V SARM1 - suggestive of more modest/moderate

**Table 1.** SARM1 coding variants in Project MinE DF1.

| SARM1 domain | Coding variant* | rsID | Chr.17 position and change† | European MAF‡ | Carriers Cases (4,366) | Controls (1,832) |
|---|---|---|---|---|---|---|
|  | T39M | rs988047470 | 28372148 C > T | 0 |  | 1 |
|  | L76fs | - | 28372255 del | 0 |  | 1 |
|  | V88M | rs1555584160 | 28372294 G > A | 0 |  | 1 |
|  | V112I | rs1032963037 | 28372366 G > A | 0.000015 | 1 |  |
|  | L223P | - | 28381400 T > C | 0 | 1 |  |
|  | Δ229–235 § | rs782325355 | 28381417_37 del | 0.000059 | 2 |  |
|  | A240E | rs1449836804 | 28381451 C > A | 0.000015 | 1 |  |
|  | R244S | - | 28381462 C > A | 0 | 1 |  |
|  | Δ249–252 | - | 28381475_86 del | 0 | 1 |  |
|  | A250T | rs1555585243 | 28381480 G > A | 0 | 1 |  |
|  | A275V | rs376587698 | 28381556 C > T | 0.000029 | 1 |  |
| ARM | A301S | rs782606059 | 28381633 G > T | 0.00012 | 3 |  |
|  | R310H | rs369186722 | 28381661 G > A | 0.00016 | 1 |  |
|  | V331E | rs1555585331 | 28381724 T > A | 0 | 1 |  |
|  | P332Q | rs140811640 | 28381727 C > A | 0.012 | 107 | 47 |
|  | N337D | rs375690432 | 28381741 A > G | 0.00038 | 3 | 1 |
|  | E340K | rs781854217 | 28381750 G > A | 0.000059 | 1 |  |
|  | A341V | rs373458416 | 28381754 C > T | 0.00012 | 2 |  |
|  | T385A | rs2068039068 | 28384420 A > G | 0 | 1 |  |
|  | Q418H | rs1194545365 | 28384521 G > T | 0.000029 | 1 | 1 |
|  | Y429F | - | 28384553 A > T | 0 |  | 1 |
|  | E431G | rs1555585662 | 28384559 A > G | 0 | 1 |  |
|  | R465T | - | 28385039 G > C | 0 | 1 |  |
|  | C482Y | - | 28385090 G > A | 0 |  | 1 |
| SAM | R484C | rs1555585809 | 28385095 C > T | 0 | 1 |  |
|  | A488E | rs782228906 | 28385108 C > A | 0.00012 | 2 |  |
|  | Y501H | rs144613221 | 28385146 T > C | 0.0029 | 32 | 13 |
|  | T50211P | rs782421919 | 28385149 A > C | 0.000074 | 1 |  |
|  | V518L | rs782106973 | 28385197 G > C | 0.000088 | 3 |  |
|  | S558N | - | 28388216 G > A | 0 |  | 1 |
|  | R569C | rs571724138 | 28388248 C > T | 0.000029 | 1 |  |
|  | R615H | rs782753946 | 28388460 G > A | 0.000044 |  | 1 |
|  | G624* | rs141324431 | 28388486 G > T | 0.000073 |  | 1 |
| TIR | D637Y | rs1451417529 | 28388525 G > T | 0 | 1 |  |
|  | A646S | rs782676389 | 28395917 G > T | 0 | 1 |  |
|  | Q673* | - | 28395998 C > T | 0 | 1 |  |
|  | A719V | rs146812537 | 28396267 C > T | 0.00050 |  | 2 |
|  | G722V | rs1298702390 | 28396276 G > T | 0 | 1 | 1 |

All ARM domain and other N-terminal variants, barring L76fs, have been tested for function in this study.

An additional frameshift variant (corresponding to 28372175dup, G46fs, rs11437592) was reported, but it appears to be an artefact due to sequence discrepancies in the GRCh37 genome build.

*numbering based on the canonical 724 amino acid human SARM1.

†numbering according to GRCh38 after lift over from GRCh37 which contains sequence discrepancies that result in anomalous numbering (see Figure 1 legend and Materials and Methods).

‡minor allele frequency for European (non-Finnish) population in gnomAD v3.1.1 as the best ethnicity match for this dataset.

§This variant is reported as Δ226–232 elsewhere (**Bloom et al., 2021**). At the DNA level the deletion encompasses a 32 bp region with 11 bp identical repeats at each end and, as such, any 21 bp deletion within this region will have an identical effect at both the DNA and protein level. Δ226–232 and Δ229–235 are thus identical proteins. We have used the Δ229–235 nomenclature for consistency with its rsID.

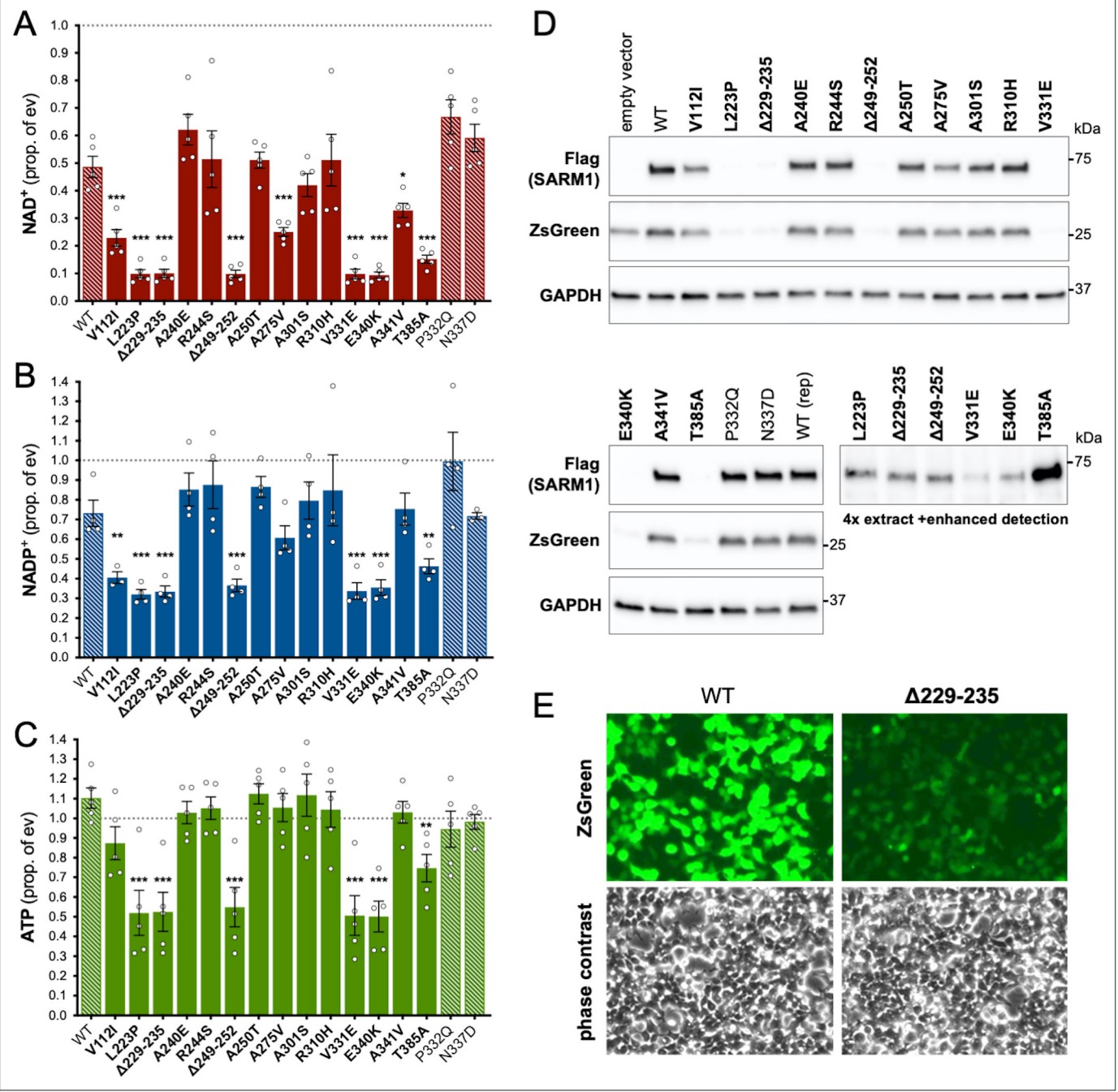

**Figure 2.** Several ALS patient-specific SARM1 ARM domain coding variants in Project MinE increase NAD$^+$ depletion in transfected HEK 293T cells. (A–C) Levels of NAD$^+$ (A), NADP$^+$ (B) and ATP (C) in extracts of HEK 293T cells 24 hr after transfection with expression constructs for Flag-tagged WT or variant SARM1 (as listed). 50 ng SARM1 vector combined with 750 ng empty vector (24-well plate format) was transfected for optimal assay conditions (see *Figure 2—figure supplement 1*). Data are shown as a proportion of levels in cells transfected with empty vector (ev) alone performed in parallel (set at 1, grey dashed line). Means ± SEM with individual data points are plotted (n = 4 or 5). *p < 0.05, **p < 0.01, and ***p < 0.001, multiple pairwise comparisons to WT SARM1 with FDR correction (after log transformation of data in parts A and B). Bars for WT SARM1, and the P332Q and N337D SARM1 variants, which are seen in both patient and control groups in Project MinE, are hatched to differentiate them from patient-specific variants (bold text, filled bars). (D) Representative immunoblots of the extracts described in parts A-C showing Flag-tagged WT or variant SARM1, ZsGreen (co-expressed with the exogenous SARM1 from a bicistronic mRNA) and GAPDH (acting as a loading control). Molecular weight markers (kDa) are shown on the right of each panel. Four times the amount of extract and enhanced detection was needed to detect SARM1 variants that cause the greatest NAD$^+$ depletion (bottom right blot). (E) Representative ZsGeen fluorescence and phase contrast imaging of HEK 293T cells 24 hr after transfection with WT or Δ229–235 SARM1 expression constructs (brightness and contrast have been increased equally in each pair of images to enhance visualisation). Reduced ZsGreen signal intensity, as seen here for Δ229–235 SARM1, was consistently seen (n = 5) for variants that reduced NAD$^+$ levels more than WT SARM1.

*Figure 2 continued on next page*

*Figure 2 continued*

The online version of this article includes the following figure supplement(s) for figure 2:

**Source data 1.** Source data for *Figure 2*, *Figure 2—figure supplements 1 and 2*.

**Figure supplement 1.** Optimisation of transfection conditions for assessing SARM1-dependent NAD$^+$ consumption in HEK 293T cells.

**Figure supplement 2.** Truncated forms of putative strong GoF SARM1 variants are not detected in transfected HEK 293T cells and supressed expression of the full-length proteins and co-expressed zsGreen is increased by boosting NAD$^+$ production with NR.

**Figure supplement 3.** Cell density-dependent death of transfected HEK 293T cells expressing SAM-TIR fragment.

GoF. All other variants depleted NAD$^+$ to a similar extent as WT SARM1. The variants that had the strongest effect on NAD$^+$ levels also caused greater depletion of NADP$^+$ and ATP than WT SARM1 (*Figure 2B,C*), with NADP$^+$ representing an alternative substrate of SARM1 glycohydrolase activity (*Essuman et al., 2018*; *Horsefield et al., 2019*; *Wan et al., 2019*) and ATP loss likely to be a lagging consequence of NAD$^+$ loss.

Importantly, the lower level of NAD$^+$ seen for some variants in these assays was not a result of greater variant overexpression. Instead, we found that the variants that caused the greatest NAD$^+$ depletion are barely detectable by immunoblotting (*Figure 2D*). By using antibodies against either the C-terminal Flag tag or the N-terminus of SARM1, we can rule out any contribution of truncated forms of the variants (*Figure 2—figure supplement 2A*). This indicates that enhanced NAD$^+$ depletion is achieved by substantially lower levels of full-length versions of some of the variants and suggests very strong GoF. IRES-driven expression of ZsGreen (in a bicistronic cassette with SARM1) was also found to be substantially lower in cells with the greatest NAD$^+$ depletion (*Figure 2D*). Crucially, fluorescence imaging suggested that this reflected suppressed ZsGreen expression within cells rather than reduced transfection efficiency (*Figure 2E*), which would anyhow be incompatible with almost complete NAD$^+$ depletion if many cells remained untransfected. However, this appears to be consistent with a previous report showing that overexpression of (WT) SARM1 suppresses *de novo* synthesis of exogenous proteins in an NADase-dependent manner (*Izadifar et al., 2021*). In support of this mechanism, we found that the cell permeable NAD$^+$ precursor nicotinamide riboside (NR) allows higher levels of expression of both zsGreen and the putative strong NADase GoF SARM1 variants themselves, presumably by boosting NAD$^+$ synthesis and thus delaying the point at which NAD$^+$ becomes limiting for *de novo* protein synthesis as a result of enhanced SARM1 NADase activity (*Figure 2—figure supplement 2B,C*). Even so, we cannot exclude reduced stability as also contributing to the low expression levels of these variants.

Previously, reduced cell viability has been reported in transfected HEK cells expressing active SARM1 mutants (*Panneerselvam et al., 2012*; *Gerdts et al., 2013*; *Sporny et al., 2019*; *Sporny et al., 2020*), but we did not see any obvious change in cell morphology suggestive of a loss of cell viability in our assays (*Figure 2E*), even when NAD$^+$ was substantially depleted. While this probably partly reflects that we use a lower amount of SARM1 expression construct in our assays, we also found that SARM1 activity-dependent loss of viability of transfected HEK 293T cells is very dependent on cell density, with the high cell densities we routinely use also conferring substantial protection (*Figure 2—figure supplement 3*).

## Project MinE DF1 putative GoF SARM1 variants have very high constitutive NADase activity

Our measurements of NAD$^+$ levels in transfected HEK 293T cells indirectly pointed to variable degrees of NADase GoF for a number of the Project MinE patient-specific SARM1 ARM domain coding variants. Therefore, we next directly tested the NADase activity of purified recombinant WT and variant SARM1. Recombinant proteins were purified from transfected HEK 293T cell lysates by immunoprecipitation following NR supplementation of the HEK 293T cell medium to boost expression and maximise yield, particularly of the strong GoF variants (see above) (*Figure 3—figure supplement 1*).

Constitutive (basal) NADase activity of the purified proteins in reactions containing just NAD$^+$ was assayed first. We used 25 µM NAD$^+$, which is close to the Km for the SARM1 catalytic site (*Essuman et al., 2018*; *Sporny et al., 2020*; *Angeletti et al., 2021*) and obtained an NAD$^+$ consumption rate for WT SARM1 (~1 mol/min/mol enzyme) comparable to that reported in a previous study (*Zhao et al., 2019*). Importantly, these assays confirmed the hypothesised NADase GoF for most of the variants

that increased depletion of NAD$^+$ in transfected HEK 293T cells, but in some cases, their constitutive activity was unexpectedly high: rates for five of the patient-specific variants - L223P, Δ229–235, Δ249–252, V331E and E340K - were found to be up to 20 times higher than WT SARM1 and a sixth - T385A - around 10 times higher (*Figure 3A*). In fact, these very high per mol constitutive NADase activities are similar to that of SARM1 lacking the extreme N-terminus and auto-inhibitory ARM domain (SAM-TIR fragment, amino acids 409–724) (*Figure 3B*).

Of the three other variants that increased NAD$^+$ depletion in transfected HEK 293T cells, V112I SARM1 has a rate a little more than twice as high as WT SARM1, whereas the activities of A275V and A341V SARM1 are not significantly raised (*Figure 3A*), although we cannot rule out subtle differences in the activities of the recombinant proteins compared to their activities in cells. Interestingly, the constitutive NADase rates calculated for P332Q and N337D, the variants present in both the patient and control groups, and R310H, a patient-specific variant, suggest partial LoF (*Figure 3A*).

## Project MinE DF1 SARM1 variants with high constitutive NADase activity are not further activated by NMN

The NADase activity of full-length SARM1 can be induced by NMN (*Zhao et al., 2019*; *Loreto et al., 2021*; *Figley et al., 2021*), a nucleotide that plays a critical role in triggering axon degeneration (*Di Stefano et al., 2015*; *Di Stefano et al., 2017*). Greater activation by NMN could thus represent an alternative pro-degenerative GoF mechanism. We therefore tested the recombinant SARM1 ARM domain coding variants for NMN-dependent triggering of their NADase activity.

We first assessed activation by 50 μM NMN. This is a higher level of NMN than would likely occur in neurons and/or axons, even after injury (*Angeletti et al., 2021*), and should thus promote robust SARM1 NADase activation in these assays. Under these conditions, the NADase activity of WT SARM1 roughly tripled but there was no clear increase for the six variants with very high constitutive activity (L223P, Δ229–235, Δ249–252, V331E, E340K, and T385A SARM1) (*Figure 3C*). This mirrors a lack of NMN-responsiveness for the SAM-TIR fragment lacking the allosteric binding site in the auto-inhibitory ARM domain (*Zhao et al., 2019* and data not shown).

In contrast, with the exception of V112I SARM1, all the variants with a constitutive activity closer to that of WT SARM1 were found to be NMN-inducible to some extent, with most showing levels of inducibility similar to WT SARM1 (*Figure 3C* and *Figure 3—figure supplement 2A,C*). P332Q SARM1, a more common variant equally abundant in ALS patients and controls, appears much more inducible than WT SARM1 (*Figure 3—figure supplement 2C*), but this largely reflects its substantially lower constitutive activity with its NMN-activated rate being only marginally higher than that of WT SARM1 (*Figure 3C* and *Figure 3—figure supplement 2A*). Indeed, none of the NMN-activated rates of any of the other NMN-responsive variants were significantly higher than that of WT SARM1, and for some (R244S, R310H and N337D SARM1) the activated rate was lower (*Figure 3C* and *Figure 3—figure supplement 2A*). On the other hand, V112I SARM1, which has moderately raised constitutive NADase activity, instead showed no NMN-responsiveness and its activity in the presence of 50 μM NMN is lower than the NMN-induced activity of WT SARM1 (*Figure 3—figure supplement 2A*).

Increased sensitivity to lower concentrations of NMN could represent an alternative pro-degenerative GoF mechanism, so we next assessed triggering of NADase activity by 10 μM NMN (excluding the non-responsive strong GoF variants). This concentration of NMN is likely to be closer to the concentration in damaged neurons / axons (*Angeletti et al., 2021*) but is the lowest concentration that gives consistent activation of WT SARM1 in these in vitro assays. The increase in activity of WT SARM1 in the presence of 10 μM NMN was roughly half that seen with 50 μM NMN (in parallel assays) and all the ALS patient variants previously found to be responsive to 50 μM NMN showed a similar or even lower relative level of activation with 10 μM NMN (*Figure 3D* and *Figure 3—figure supplement 2B,D*), apart from V112I SARM1 which again is not inducible (*Figure 3D*). As with 50 μM NMN, 10 μM NMN induced P332Q SARM1 activity to a greater extent than that of WT SARM1 but its activated rate in the presence of 10 μM NMN in this case was below that of WT SARM1 (*Figure 3D* and *Figure 3—figure supplement 2B,D*).

Together, these NADase activity data indicate that six of the exclusively patient-associated *SARM1* ARM domain alleles in Project MinE DF1 encode variants with greatly enhanced constitutive activity comparable to that of the fully disinhibited SAM-TIR fragment, and one additional patient-associated variant showed more moderate constitutive NADase GoF. Surprisingly, however, no variants showed

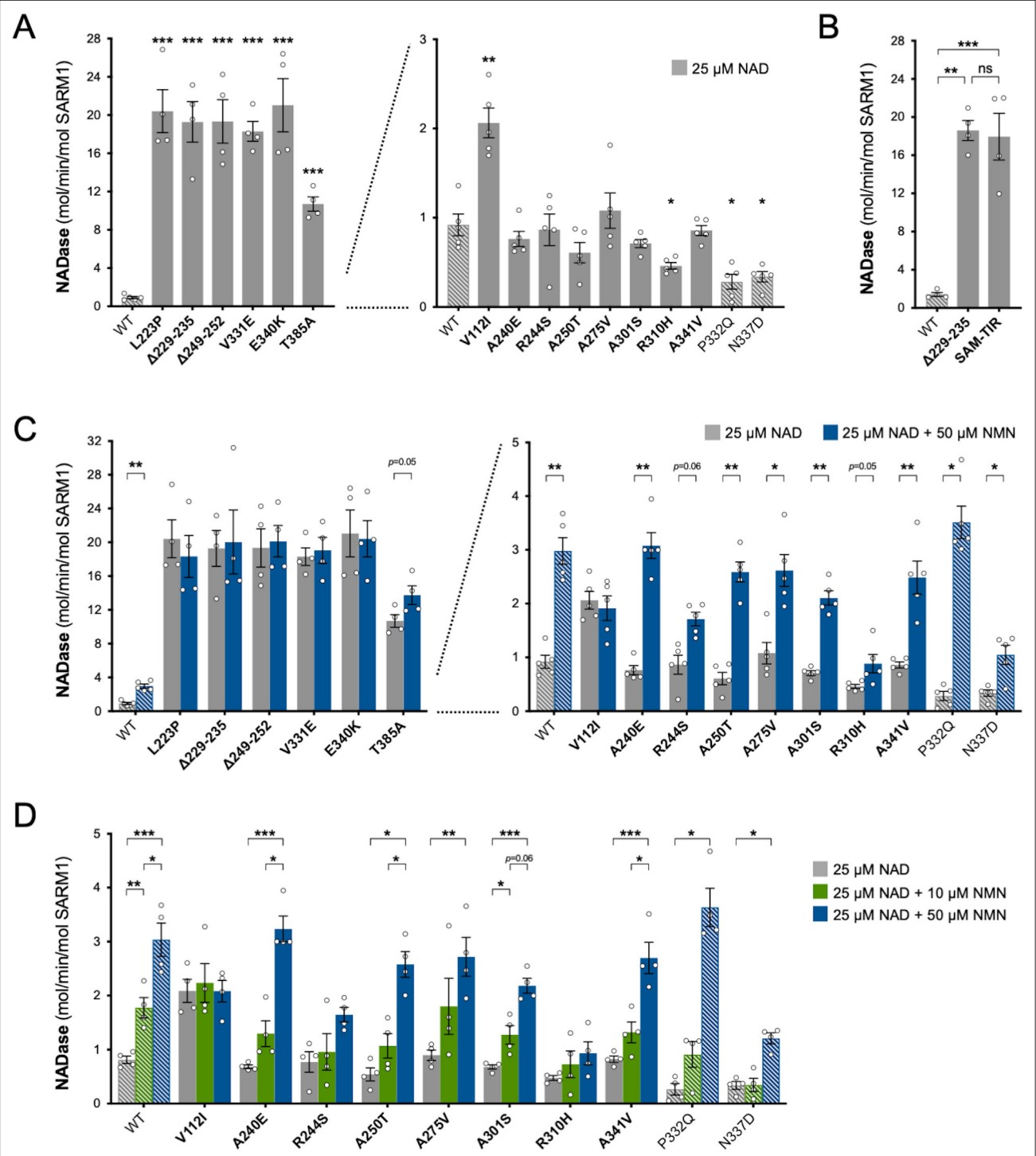

**Figure 3.** Constitutive NADase activity and NMN responsiveness of SARM1 ARM domain variants in Project MinE. Each panel is designed to show different statistical comparisons and some of the same data are used in different tests, as described. WT SARM1, and P332Q and N337D SARM1 (seen in both patient and control groups in Project MinE), are shown as hatched bars throughout to differentiate them from patient-specific variants (bold text, filled bars). (**A**) and (**B**) Constitutive (basal) NADase activities of immunoprecipitated WT SARM1, variant SARM1, or SAM-TIR fragment (as indicated) in the presence of 25 μM NAD⁺. In part A, data for variants with high or low NADase activity, obtained in parallel, are plotted separately, against different scales, to more clearly show all differences relative to WT SARM1 (included on both). Means ± SEM with individual data points (n = 4 or 5) are plotted. *p < 0.05, **p < 0.01, ***p < 0.001 and ns (not significant) = p > 0.05, multiple pairwise comparisons to WT SARM1 with FDR correction for part A and one-way ANOVA with Tukey's correction for part B (both after log transformation). (**C**) Activation of SARM1 NADase by 50 μm NMN (in the presence of 25 μM NAD⁺). Constitutive activities (no NMN) are as in part A with rates + NMN calculated from assays performed in parallel. High and low NADase

*Figure 3 continued on next page*

*Figure 3 continued*

activity variants are plotted separately (as in part A). Means ± SEM with individual data points (n = 4 or 5) are plotted. *p < 0.05 and **p < 0.01, multiple paired *t* tests of rates ± NMN with FDR correction (after log transformation). (**D**) Activation of SARM1 NADase by 10 or 50 µm NMN (in the presence of 25 µM NAD⁺). NMN-insensitive variants with very high constitutive NADase activity were not tested. Constitutive and +50 µM NMN activities are a subset of those shown in part B and are only those from assays performed in parallel with 10 µM NMN. Means ± SEM with individual data points (n = 4) are plotted. *p < 0.05, **p < 0.01, and ***p < 0.001, two-way ANOVA with Šidák's correction (after log transformation).

The online version of this article includes the following figure supplement(s) for figure 3:

**Source data 1.** Source data for *Figure 3*, *Figure 3—figure supplements 1 and 2*.

**Figure supplement 1.** Purification of recombinant SARM1 proteins for use in NADase assays.

**Figure supplement 2.** Relative NMN responsiveness of Project MinE DF1 SARM1 ARM domain variants.

enhanced responsiveness to NMN from the normal basal level. Thus, the point mutations and small in-frame deletions that confer the strongest GoF mimic the effects of complete removal of ARM domain auto-inhibition.

## Identification of an additional ALS patient-associated strong GoF SARM1 variant in the Answer ALS project database

We next extended our search for putative ALS-associated *SARM1* GoF alleles to the Answer ALS project database consisting of WGS data for 706 ALS patients and 92 matched controls (*Rothstein et al., 2020*). Some of the Project MinE DF1 variants were independently found in this new, non-overlapping dataset, including one occurrence of the strong NADase GoF Δ229–235 *SARM1* allele in a patient, but we also found three new patient-associated alleles and one new control-associated allele encoding changes within the ARM domain or just N-terminal to it (encompassing amino acids 29–397) (*Figure 1* and *Table 2*). Three additional coding variants are also present in this expanded region in control subjects in Project MinE DF1 (*Figure 1* and *Table 1*). Therefore, given the precise N-terminal boundary of the ARM domain is not well-defined, we assessed the NADase function of all these previously untested coding variants from both databases, with the exception of Project MinE DF1 frameshift variant L76fs that can reasonably be assumed to be a full LoF allele.

Using the same assays as for the original Project MinE variants, we identified patient-associated Answer ALS variant, R267W SARM1, as an additional strong GoF, with robust NAD⁺-consuming activity in transfected HEK 293T cells combined with low expression (*Figure 4A and B*) and a constitutively

**Table 2.** SARM1 coding variants in Answer ALS.

| SARM1 domain | Coding variant* | rsID | Chr.17 position and change† | European MAF‡ | Carriers Cases (706) | Controls (92) |
|---|---|---|---|---|---|---|
| ARM | G35E | rs1480321233 | 28372136 G > A | 0.000015 | 1 | |
| | R131G | rs1389320808 | 28372423 C > G | 0.000015 | 1 | |
| | A210V | - | 28381361 C > T | 0 | | 1 |
| | Δ229–235 § | rs782325355 | 28381417_37 del | 0.000059 | 1 | |
| | R267W | rs11658194 | 28381531 C > T | 0.000015 | 1 | |
| | A301S § | rs782606059 | 28381633 G > T | 0.00012 | 1 | |
| | P332Q § | rs140811640 | 28381727 C > A | 0.012 | 16 | 3 |
| SAM | Y501H § | rs144613221 | 28385146 T > C | 0.0029 | 5 | 5 |
| TIR | S684F | rs782256561 | 28396162 C > T | 0.000044 | 1 | |
| | A719V § | rs146812537 | 28396267 C > T | 0.00050 | 1 | |

All ARM domain and other N-terminal variants have been tested for function in this study.

*numbering based on the canonical 724 amino acid human SARM1.

†numbering according to genome build GRCh38.

‡minor allele frequency for European (non-Finnish) population in gnomAD v3.1.1 as the best ethnicity match for this dataset.

§Variants also present in the Project Mine DF1 dataset.

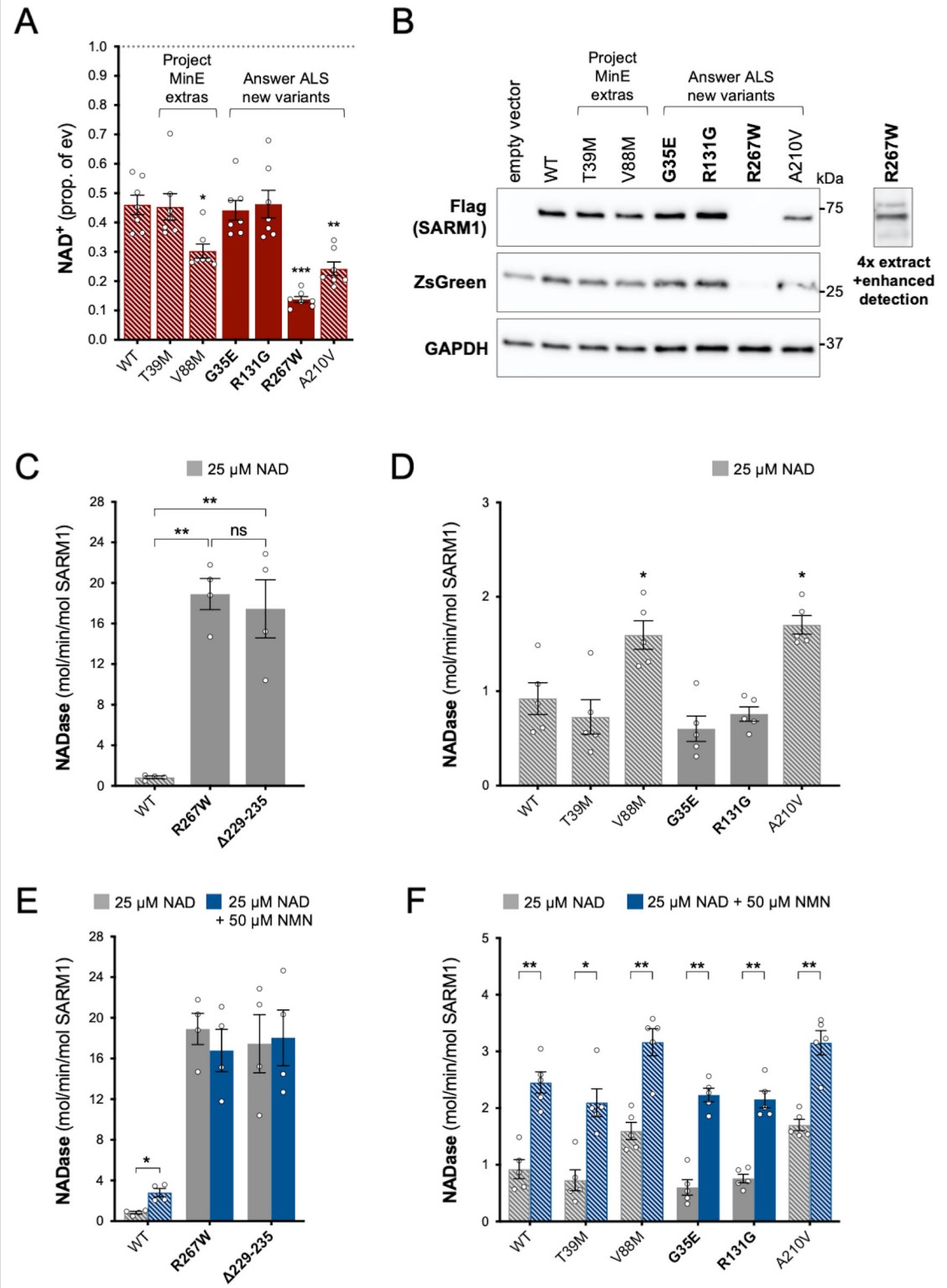

**Figure 4.** Functional characterisation of additional ARM domain and other N-terminal SARM1 coding variants in the Answer ALS database and Project MinE DF1. Bars for WT and variants that are only seen in controls are hatched throughout to differentiate them from patient-specific variants (bold text, filled bars). (**A**) NAD$^+$ levels in extracts of HEK 293T cells 24 hr after transfection with expression constructs for Flag-tagged WT or variant SARM1 (as listed). Details are as per *Figure 2*. Means ± SEM with individual data points (n = 7) are plotted. *p < 0.05, **p < 0.01, and ***p < 0.001, multiple pairwise comparisons to WT SARM1 with FDR correction (after log transformation). (**B**) Representative immunoblots of extracts as described in part A. Blots were probed as in *Figure 2D*. Four times the amount of extract and enhanced detection was needed to detect R267W SARM1 (right). Molecular

*Figure 4 continued on next page*

*Figure 4 continued*

weight markers (kDa) are shown on the right of each panel. (**C**) and (**D**) Constitutive NADase activities (at 25 µM NAD$^+$) of recombinant WT and variant SARM1, as indicated. Predicted strong GoF R267W SARM1 was compared directly to WT SARM1 and strong NADase GoF Δ229–235 SARM1 in part C, separately from the other variants in part D. Means ± SEM with individual data points (n = 4 or 5) are plotted. *p < 0.05, **p < 0.01, ***p < 0.001 and ns (not significant) = p > 0.05, one-way ANOVA with Tukey's correction (after log transformation) in part C and multiple pairwise comparisons to WT SARM1 with FDR correction in part D. (**E**) and (**F**) Activation of the NADase activity of recombinant WT and variant SARM1, as indicated, by 50 µm NMN (in the presence of 25 µM NAD$^+$). Constitutive activities (no NMN) are as in parts C and D with rates + NMN calculated from assays performed in parallel. Means ± SEM with individual data points (n = 4 or 5) are plotted. *p < 0.05 and **p < 0.01, multiple paired *t* tests of rates ± NMN with FDR correction (after log transformation in part E).

The online version of this article includes the following figure supplement(s) for figure 4:

**Source data 1.** Source data for *Figure 4*.

high, non-inducible NADase activity equivalent to that of the strongest Project MinE GoF variants (*Figure 4C,E*). Interestingly, two of the control-associated variants, V88M SARM1 (Project MinE DF1) and A210V SARM1 (Answer ALS), also reduced NAD$^+$ levels more than WT SARM1 in transfected HEK 293T cells (*Figure 4A*), but their constitutive activities in NADase assays were only modestly increased relative to WT SARM1 and far below the 10 times higher activity we define for strong GoF variants (*Figure 4D*). In addition, both show normal NMN-inducibility (*Figure 4F*). The remaining patient- and control-associated variants had properties similar to WT SARM1 or appeared to be modest LoF (*Figure 4*).

## Strong NADase GoF Δ229-235 SARM1 retains its GoF properties in complexes with WT SARM1

Given the very low levels of endogenous SARM1 in our HEK 293T cell line, our data so far will largely reflect the properties of each SARM1 variant in isolation. However, ALS patients heterozygous for *SARM1* GoF alleles will express the SARM1 variant alongside WT SARM1. Therefore, to mimic hetero-zygosity, we co-expressed WT and strong GoF SARM1 in HEK 293T cells, using Δ229–235 SARM1 as a representative strong GoF variant as it is most frequently seen in patients.

We found that NAD$^+$ depletion in HEK 293T cells co-transfected with equal amounts of WT and Δ229–235 SARM1 expression constructs was broadly similar to that in cells transfected with the Δ229–235 SARM1 construct alone (*Figure 5A*), although enzyme assays of the recombinant proteins are more effective at differentiating between moderate and high NADase GoF (see below). Expression of ZsGreen and SARM1 (both Δ229–235 and WT SARM1) was also strongly suppressed in both situations where Δ229–235 SARM1 is present in the transfected HEK 293T cells (*Figure 5B*), further supporting the model in which *de novo* synthesis of proteins is suppressed by enhanced SARM1 NADase activity.

SARM1 readily homo-multimerises in solution to form an octameric ring structure (*Gerdts et al., 2013*; *Horsefield et al., 2019*; *Sporny et al., 2019*; *Bratkowski et al., 2020*; *Jiang et al., 2020*; *Sporny et al., 2020*; *Figley et al., 2021*; *Shen et al., 2021*) and our finding that (Flag-tagged) Δ229–235 SARM1 efficiently co-immunoprecipitates with (HA-tagged) WT SARM1 from co-expressing cells is indicative that hetero-oligomers of WT and variant SARM1 are also readily formed (*Figure 5C*). Direct assaying of the NADase activity of hetero-oligomers revealed their constitutive NADase activity to be intermediate between that of each protein alone, but this still represents very strong GoF relative to WT SARM1 alone (*Figure 5D*). This is consistent with each protein retaining its own individual constitutive activity within the hetero-oligomers, although their lack of NMN-responsiveness suggests that WT SARM1 within the complexes may be altered in that respect (*Figure 5E*).

## Acute expression of strong GoF Δ229-235 SARM1 reduces viability of sympathetic neurons under stress conditions in an NADase-dependent manner

To test the effects of greatly enhanced constitutive activity in neurons, we used an established method involving microinjection of controlled amounts of expression constructs into cultured sympathetic neurons from the superior cervical ganglion (SCG neurons), a neuronal subtype very amenable to this approach (*Gilley and Loreto, 2020*).

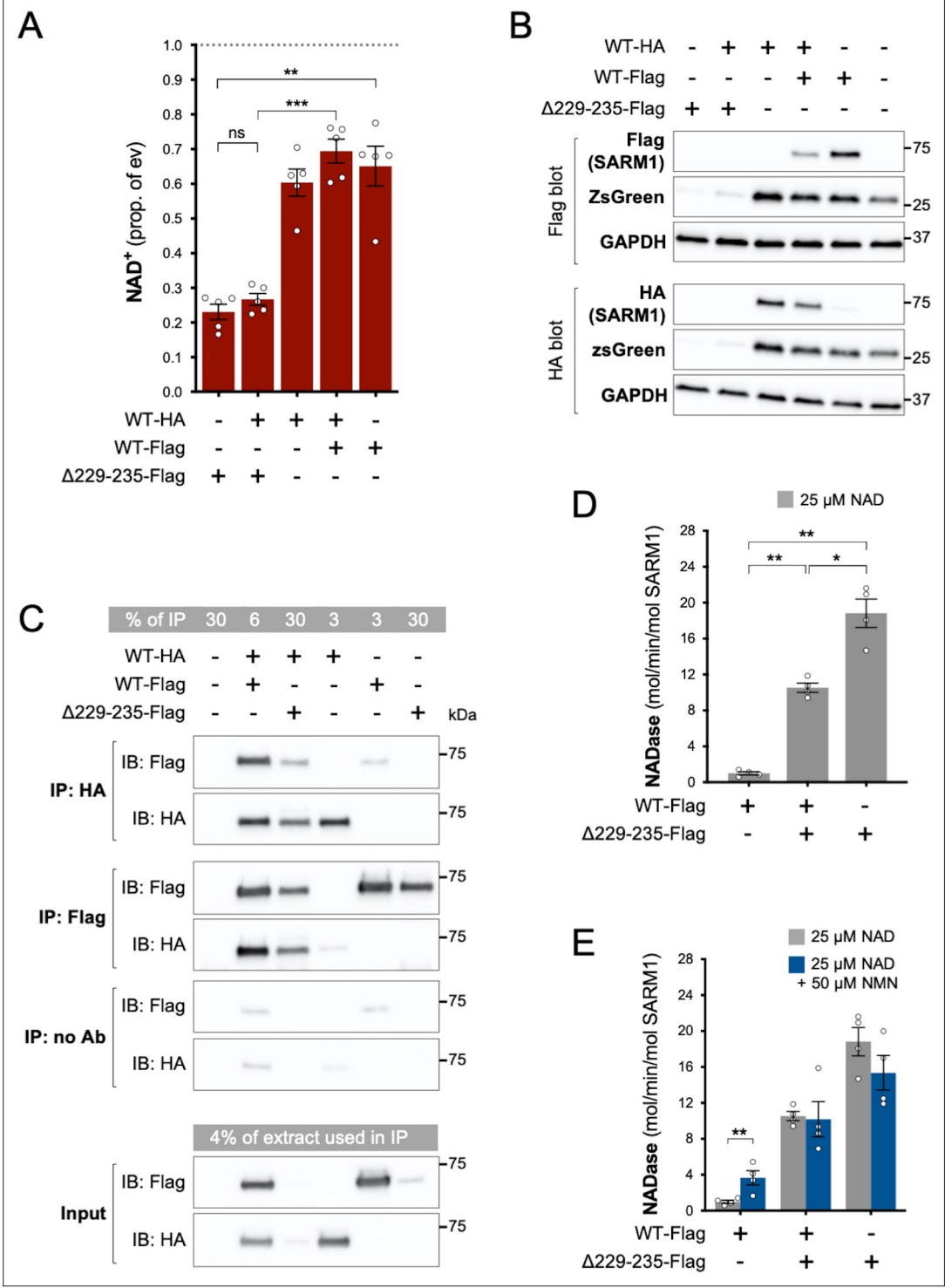

**Figure 5.** Δ229–235 SARM1 retains its strong NADase GoF properties in complexes with WT SARM1. (**A**) NAD+ levels in extracts of HEK 293T cells 24 hr after transfection with individual expression constructs or mixtures of constructs for HA- or Flag-tagged WT SARM1 and/or Flag-tagged Δ229–235 SARM1. Details as per *Figure 2*, except that 50 ng SARM1 vector was combined with 750 ng empty vector for single constructs and 25 ng each for mixtures. Means ± SEM with individual data points (n = 5) are plotted. **p < 0.01, ***p < 0.001 and ns (not significant) = p > 0.05, one-way ANOVA with Tukey's correction (after log transformation). Only relevant comparisons are shown. (**B**) Representative immunoblots of extracts as described in part A. Duplicate blots were probed with Flag or HA antibodies to independently assess levels of co-expressed Flag- or HA-tagged WT or Flag-

*Figure 5 continued*

tagged Δ229–235 SARM1. Both blots were probed for ZsGreen and GAPDH (acting as a loading control) to confirm the blots were representative replicates. A faint, non-specific cross-reaction of the HA antibody to Flag-tagged SARM1 was consistently seen under the conditions used (WT-Flag only lane). Molecular weight markers (kDa) are shown on the right of each panel. (**C**) Representative immunoblots (of n = 3) of reciprocal co-immunoprecipitation of HA-tagged WT SARM1 with Flag-tagged WT or Δ229–235 SARM1. Immunoprecipitations using Flag or HA antibodies or no antibody were performed on the same extracts of transfected HEK 293T cells co-expressing HA-tagged WT SARM1 with Flag-tagged WT or Flag-tagged Δ229–235 SARM1 (from equal amounts of construct), or expressing individual proteins. Duplicate immunoblots of immunoprecipitated proteins, and the original extracts used in the immunoprecipitations (input), were probed with Flag or HA antibodies. A small amount of non-specific binding of tagged proteins to protein A/G beads was consistently seen for inputs with the highest expression levels. Molecular weight markers (kDa) are shown on the right of each panel. (**D**) Constitutive NADase activity (in the presence of 25 μM NAD⁺) of immunoprecipitated WT + Δ229–235 SARM1 complexes compared to WT or Δ229–235 SARM1 alone. All proteins were Flag-tagged to ensure complete immunoprecipitation of complexes of WT and Δ229–235 SARM1. Means ± SEM with individual data points (n = 4) are plotted. *p < 0.05 and **p < 0.01, one-way ANOVA with Tukey's correction (after log transformation). (**E**) Inducibility of immunoprecipitated WT + Δ229–235 SARM1 complexes and WT or Δ229–235 SARM1 alone by 50 μm NMN (in the presence of 25 μM NAD⁺). Constitutive activities (no NMN) are as in part D with rates + NMN calculated from assays performed in parallel. Means ± SEM with individual data points (n = 4) are plotted. **p < 0.01, multiple paired *t* tests of rates ± NMN with FDR correction (after log transformation).

The online version of this article includes the following figure supplement(s) for figure 5:

**Source data 1.** Source data for *Figure 5*.

We first compared the effects of our representative strong GoF SARM1 variant, Δ229–235 SARM1, with those of WT SARM1. We used wild-type SCG neurons for these experiments so that exogenous Δ229–235 SARM1 would be expressed alongside endogenous (WT) SARM1 in order to partially model neurons in individuals heterozygous for the Δ229–235 *SARM1* allele. However, we minimised the amount of SARM1 expression vector injected to reduce possible artefacts of high expression, whilst still maintaining levels that are functionally relevant (*Figure 6—figure supplement 1*), so the endogenous murine SARM1 will likely be in excess of the exogenous SARM1 in these experiments. A DsRed expression construct was co-injected to improve visualisation of the targeted neurons and their neurites (co-expression of ZsGreen from the SARM1 construct being insufficient under these conditions) and we consistently saw substantially fewer DsRed labelled neurons 24 hr after injection with the Δ229–235 SARM1 construct compared to the WT SARM1 construct or empty vector (*Figure 6A,B* and *Figure 6—figure supplement 2A*). The intensity of DsRed in the few labelled Δ229–235 SARM1 neurons that were present was also greatly reduced. We found that NR supplementation boosted DsRed expression and visualisation of the Δ229–235 SARM1 neurons (*Figure 6A,B*) suggesting that strong NADase GoF suppresses DsRed expression in neurons in the same way that it suppresses co-expression of ZsGreen in transfected HEK 293T cells (*Figure 2D,E* and *Figure 2—figure supplement 2B,C*), although a separate effect on neuronal viability also cannot be ruled out.

The greatly improved visualisation of neurons injected with the Δ229–235 SARM1 expression construct in the presence of NR allowed us to next assess their response to various stresses. Intriguingly, we found that the modest stresses associated with changes in culture conditions as a result of media change, even to new NR-containing medium, was sufficient to trigger a substantial loss of the Δ229–235 SARM1 neurons (compared to not changing media), and further manipulations designed to reduce NAD⁺ biosynthesis – removal of NR, or a combination of NR removal and incubation with NAMPT inhibitor FK866 to block the rate-limiting step in the NAD⁺ salvage pathway – caused additional, incremental decreases in cell survival (*Figure 6C and D*). In contrast, neurons injected with the WT SARM1 construct or empty vector were not substantially affected by any of the manipulations (*Figure 6C and D* and *Figure 6—figure supplement 2*). Importantly, we could not differentiate the timing of cell death and neurite degeneration in these experiments, suggesting either that the primary effect is on neuronal viability or that separate cell body and neurite death mechanisms occur simultaneously.

In order to assess whether NADase activity is required for the pro-death effects of Δ229–235 SARM1, we combined the Δ229–235 deletion with the E642A NADase-inactivating mutation (*Essuman et al., 2017*). We found that double mutant Δ229–235/E642A SARM1, like single mutant E642A SARM1, has

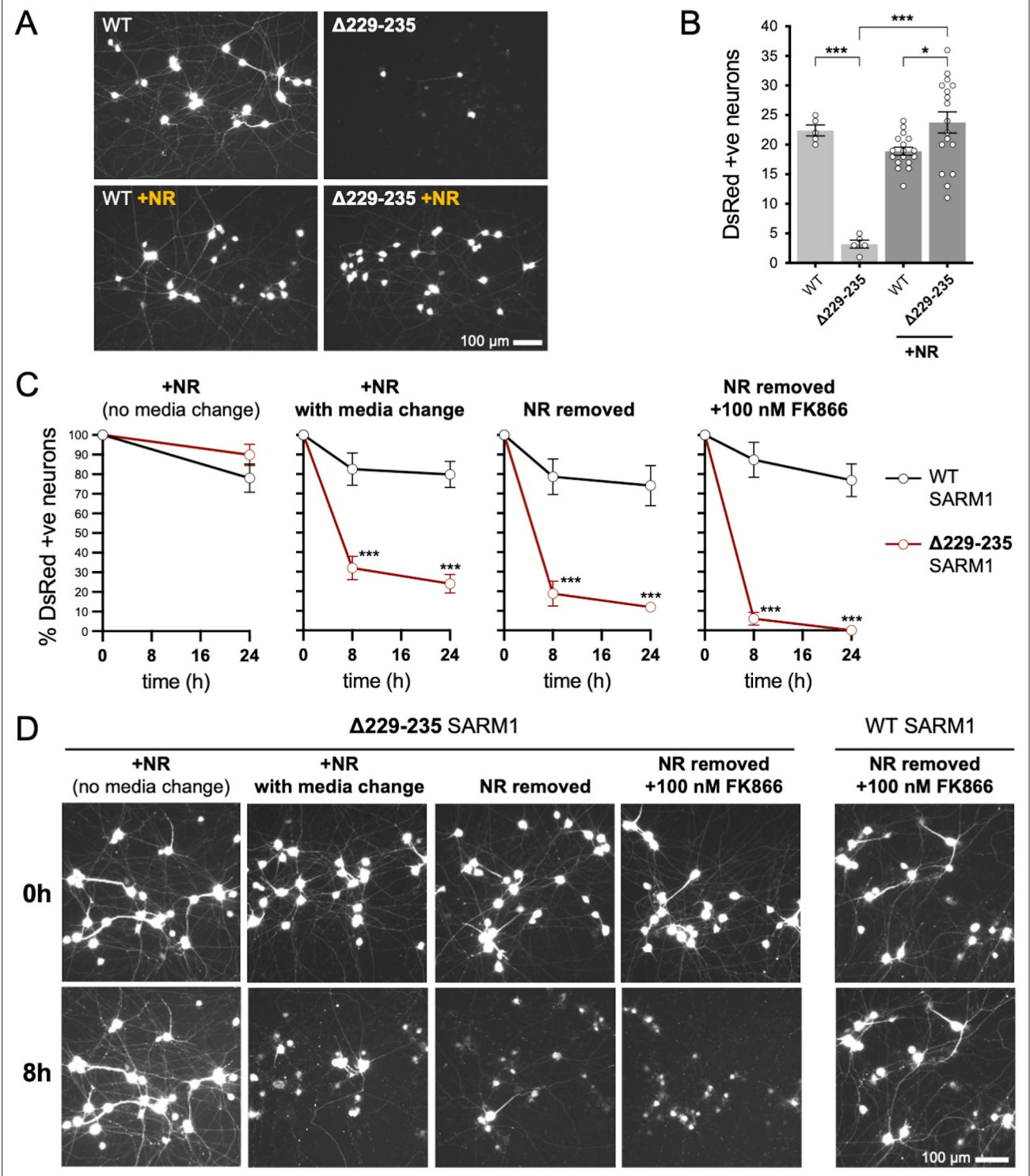

**Figure 6.** Strong NADase GoF Δ229–235 SARM1 reduces SCG neuron viability under mild stress conditions. (**A**) Representative images of DsRed-positive wild-type SCG neurons 24 hr after microinjection with expression constructs for WT or Δ229–235 SARM1 and DsRed (at 2.5 ng/μl and 40 ng/μl in the injection mix, respectively), with or without supplementation of the culture medium with 2 mM NR at the time of injection. Equal numbers of neurons were injected. (**B**) Quantification of DsRed-positive neurons for the conditions described in part A. Means ± SEM with individual data points (n = 5 or 17) are plotted. *p < 0.05 and ***p < 0.001, one-way ANOVA with Tukey's correction. Only relevant comparisons are shown. (**C**) Time course of survival of DsRed-positive neurons after manipulation of culture conditions as indicated. Neurons were injected as in part A and maintained in medium containing NR. Manipulations were performed 24 hr after injection which corresponds to time 0 on the graphs. Survival of injected, DsRed-labelled neurons was assessed as described in part D and is expressed as a percentage of those present at time 0. Means ± SEM (n = 4 or 5) are plotted. ***p <

*Figure 6 continued on next page*

*Figure 6 continued*

0.001, two-way ANOVA with Šidák's correction. (**D**) Representative images for data presented in part C. Only one representative condition is shown for neurons injected with the WT SARM1 construct. Survival of DsRed-positive neurons was assessed by comparing signal intensity and neuron size / shape at different timepoints, a method that provides an accurate reflection of survival rates (*Gilley and Coleman, 2010*).

The online version of this article includes the following figure supplement(s) for figure 6:

**Source data 1.** Source data for *Figure 6*, *Figure 6—figure supplements 1 and 2*.

**Figure supplement 1.** Microinjection of 2.5 ng/µl WT SARM1 expression construct provides expression that is functionally relevant.

**Figure supplement 2.** Injection of empty vector does not affect SCG neuron viability.

little or no effect on NAD$^+$ levels in transfected HEK 293T cells (*Figure 7A*), consistent with a lack of NADase activity. We also saw much more robust expression of Δ229–235/E642A SARM1 and co-expressed ZsGreen compared to cells expressing Δ229–235 SARM1 (*Figure 7B*). This further supports a correlation between SARM1 NADase activity and *de novo* protein synthesis. Crucially, in contrast to Δ229–235 SARM1, SCG neurons injected with a Δ229–235/E642A SARM1 construct were easily identified, even in the absence of NR supplementation, and were not sensitive to combined NR removal and incubation with NAMPT inhibitor FK866 (after being cultured in the presence of NR) (*Figure 7C and D*). Therefore, NADase activity is required for the pro-death effects of Δ229–235 SARM1 in this assay.

## Evidence for enrichment of strong GoF SARM1 variants in ALS, HSP and other motor nerve disorder patients

Among the total of 23 SARM1 ARM domain and other N-terminal coding variants seen in patients and controls in the Project MinE DF1 and Answer ALS databases, we have identified a group of seven of the exclusively patient-associated variants – L223P, Δ229–235, Δ249–252, R267W, V331E, E340K and T385A SARM1 – as having very strong constitutive NADase GoF, with each being 10–20 times more active than WT SARM1. No other variants came close to this level, although V112I (in a patient) and V88M and A210V (in controls) showed more modest increases in constitutive activity to 1.7–2.2 times that of WT SARM1. Therefore, within these datasets, strong NADase GoF *SARM1* alleles are only seen in patients and never in controls. This is consistent with our hypothesis that strong SARM1 NADase GoF is a risk factor for ALS.

In order to investigate this association in more depth, we assessed whether any of the 23 variants assessed for function in this study appear in other relevant cohorts, including cohorts containing patients with HSP and other motor nerve disorders. If the association is robust, then the rare, strong GoF variant alleles identified here should remain enriched in patients and underrepresented in control individuals, while variants with properties more like WT SARM1 would not show consistent association with either group. We identified the following additional datasets for this purpose: case and control samples in Project MinE data freeze 2 (DF2) that were not part of DF1; samples from motor nerve disorder cases and selected control groups (see Materials and methods) from the GENESIS project (*Gonzalez et al., 2015*) and a UCL rare disease (neurology) dataset; cases and controls from an HSP study; and the Lothian birth cohort (LBC) (http://www.lothianbirthcohort.ed.ac.uk) as an additional aged control cohort. Among these datasets, four additional cases are heterozygous for one of the strong GoF alleles (two ALS cases, one HSP case and one upper and lower motor nerve disorder case) whereas no controls in any dataset were identified as carriers of these alleles (*Figure 8*). Therefore, in total, 13 of the 11,117 patients in the combined datasets are heterozygous for a strong GoF *SARM1* allele (13 out of 22,234 alleles), with no carriers among 10,402 controls (0 out of 20,804 alleles) (*Figure 8*). In contrast, a number of the other alleles encoding variants with properties more similar to WT SARM1, including some of those previously only associated with cases in Project MinE DF1 and Answer ALS, were present in both cases and controls in the new datasets (*Figure 8*).

We next performed statistical analyses on aggregated data from all the datasets to test the strength of the association specifically between SARM1 GoF and motor nerve disorders. Crucially, burden testing of the group of seven strong GoF variant alleles together in the combined datasets revealed a highly significant association with disease (burden test and SKAT-O p = 0.00017), whereas there was no clear association with disease when considering all the other alleles as a group (burden test p = 0.47, SKAT-O p = 0.61). In addition, there is a greater than four times enrichment of

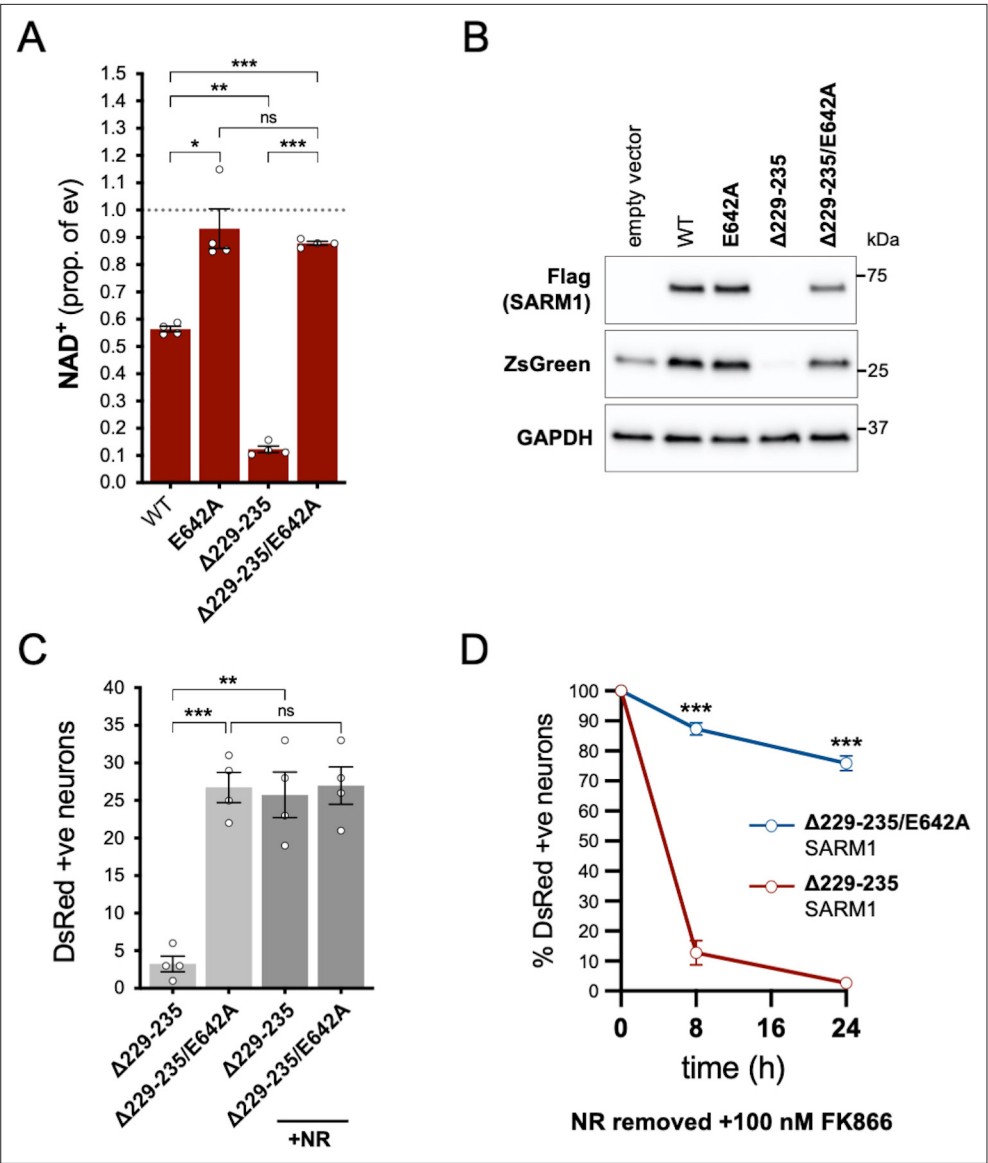

**Figure 7.** NADase activity is required for the pro-death effects of Δ229–235 SARM1 in SCG neurons. (**A**) Relative levels of NAD⁺ in extracts of HEK 293T cells 24 hr after transfection with expression constructs for WT, E642A, Δ229–235 or Δ229–235/E642A SARM1, as per **Figure 2**. Means ± SEM with individual data points (n = 4) are plotted. *p < 0.05, **p < 0.01, ***p < 0.001 and ns (not significant) = p > 0.05, one-way ANOVA with Tukey's correction (after log transformation). Only relevant comparisons are shown. (**B**) Representative immunoblots of extracts as described in part A. Blots were probed as in **Figure 2D**. Molecular weight markers (kDa) are shown on the right of each panel. (**C**) Numbers of DsRed-positive neurons 24 hr after microinjection with expression constructs for Δ229–235 SARM1 or Δ229–235/E642A SARM1 and DsRed (at 2.5 ng/μl and 40 ng/μl in the injection mix, respectively), with or without supplementation of the culture medium with 2 mM NR at the time of injection. Equal numbers of neurons were injected. Means ± SEM with individual data points (n = 4) are plotted. **p < 0.01, ***p < 0.001 and ns = p > 0.05, one-way ANOVA with Tukey's correction. Only relevant comparisons are shown. (**D**) Time course of survival of DsRed-positive neurons injected as in part C and maintained in medium containing NR prior to removal of NR and addition of 100 nM FK866 24 hr after injection (corresponding to time 0 on the graphs). Survival of injected neurons was assessed as described in **Figure 5D** and is expressed as a percentage of those present at time 0. Means ± SEM (n = 4) are plotted. ***p < 0.001, two-way ANOVA with Šidák's correction.

The online version of this article includes the following figure supplement(s) for figure 7:

**Source data 1.** Source data for **Figure 7**.

| | Total alleles | Strong GoF | | | | | | | | Other | | | | | | | | | | | | | | | | P332Q† |
|---|---|---|---|---|---|---|---|---|---|---|---|---|---|---|---|---|---|---|---|---|---|---|---|---|---|---|
| | | L223P | Δ229-235 | Δ249-252 | R267W | V331E | E340K | T385A | Combined | G35E | T39M | L76fs | V88M | V112I | R131G | A210V | A240E | R244S | A250T | A275V | A301S | R310H | N337D | A341V | Combined | |
| **Cases** MinE DF1 | 8,732 | 1 | 2 | 1 | - | 1 | 1 | 1 | 7 | - | - | - | - | 1 | - | - | 1 | 1 | 1 | 1 | 3 | 1 | 3 | 2 | 14 | 107 |
| Answer ALS | 1,412 | - | 1 | - | 1 | - | - | - | 2 | 1 | - | - | - | - | 1 | - | - | - | - | - | 1 | - | - | - | 3 | 16 |
| MinE DF2+* | 3,176 | - | - | - | - | - | - | 1 | 1 | - | - | - | - | - | - | - | - | - | - | - | - | 1 | - | - | 1 | 40 |
| GENESIS | 5,936 | - | 2 | - | - | - | - | - | 2 | - | - | - | - | - | - | - | - | - | - | - | - | - | - | 1 | 1 | 15 |
| UCL | 1,366 | - | - | - | 1 | - | - | - | 1 | - | - | - | - | - | - | - | - | - | - | - | 2 | - | - | - | 2 | 5 |
| HSP study | 1,612 | - | - | - | - | - | - | - | 0 | - | - | - | - | - | - | - | - | - | - | - | - | - | - | - | 0 | 15 |
| **Total** | **22,234** | 1 | 5 | 1 | 2 | 1 | 1 | 2 | **13** | 1 | 0 | 0 | 0 | 1 | 1 | 0 | 1 | 1 | 1 | 1 | 6 | 2 | 3 | 3 | **21** | 198 |
| **Controls** MinE DF1 | 3,664 | - | - | - | - | - | - | - | 0 | - | 1 | 1 | 1 | - | - | - | - | - | - | - | - | - | 1 | - | 4 | 47 |
| Answer ALS | 184 | - | - | - | - | - | - | - | 0 | - | - | - | - | - | - | 1 | - | - | - | - | - | - | - | - | 1 | 3 |
| MinE DF2+* | 812 | - | - | - | - | - | - | - | 0 | - | - | - | - | - | - | - | - | - | - | - | - | - | - | 1 | 1 | 15 |
| GENESIS | 7,330 | - | - | - | - | - | - | - | 0 | - | - | - | - | - | - | - | - | - | - | 1 | - | - | 1 | 1 | 3 | 58 |
| UCL | 3,616 | - | - | - | - | - | - | - | 0 | - | - | - | - | - | - | - | - | - | - | 1 | 1 | 2 | - | - | 4 | 38 |
| HSP study | 2,490 | - | - | - | - | - | - | - | 0 | - | - | - | - | - | - | - | - | - | - | - | - | - | - | - | 0 | 19 |
| LBC | 2,708 | - | - | - | - | - | - | - | 0 | - | - | - | - | - | - | - | - | - | - | - | - | - | 2 | - | 2 | 26 |
| **Total** | **20,804** | 0 | 0 | 0 | 0 | 0 | 0 | 0 | **0** | 0 | 1 | 1 | 1 | 0 | 0 | 1 | 0 | 0 | 0 | 2 | 1 | 2 | 4 | 2 | **15** | 206 |

**Figure 8.** Occurrences of SARM1 alleles encoding ARM domain and other N-terminal variants in individual datasets. * Extra samples in Project MinE DF2 that were not part of DF1. † MAFs for the P332Q SARM1 allele in each dataset (in cases plus controls combined) of 0.012 (MinE DF1), 0.012 (Answer ALS), 0.014 (MinE DF2 extras), 0.0055 (GENESIS), 0.0086 (UCL), 0.0010 (LBC) and 0.0083 (HSP study) are, barring the more ethnically-diverse GENESIS cohorts, broadly consistent with each dataset containing predominantly European samples as the European MAF of 0.012 is substantially higher than in other populations (gnomAD v3.1.1).

The online version of this article includes the following source data for figure 8:

**Source data 1.** Source data for *Figure 8*.

the strong GoF alleles among our aggregated patient cohorts compared to European (non-Finnish) non-neuro samples from gnomAD v3.1.1 (*Table 3* - MAF of 0.00058 vs 0.00014) and, despite the presence of a few carriers of strong GoF *SARM1* alleles in this gnomAD cohort and the uncertainty regarding their health status (see *Table 3* legend), their inclusion as additional controls in the statistical tests actually strengthens the association between the *SARM1* strong GoF alleles and disease (burden test p = 0.000025, SKAT-O p = 0.000057). Notably, disease-association also remains even when the patient-associated V112I *SARM1* allele and control-associated V88M and A210V *SARM1* alleles, encoding more modest GoF variants, are grouped with the strong GoF alleles (burden test p = 0.0042, SKAT-O p = 0.0021 excluding gnomAD samples, and burden test p = 0.000047, SKAT-O p = 0.00019 including gnomAD samples). Together, these tests suggest that SARM1 ARM domain coding variation that results in constitutive NADase hyperactivation is very strongly associated with ALS and other motor nerve disorders. Individual-level data were not available for all datasets to identify covariates to correct for population stratification, but the use of broadly matched datasets to minimise the effect of population variation (see Materials and methods) means that results are likely to reflect true disease-associated genetic variation.

## Phenotypic heterogeneity and variable age-of-onset in patients with *SARM1* GoF alleles

Finally, we collated clinical information for all the individuals in our combined datasets that are heterozygous for the *SARM1* alleles encoding the strong GoF variants identified in this study (*Table 4*). Information is also provided for one additional ALS case heterozygous for the Δ229–235 variant *SARM1* allele from an independent source, and for the one individual heterozygous for the allele encoding moderate GoF V112I SARM1 for comparison. Intriguingly, as well as confirming there is no patient

**Table 3.** Occurrences and MAFs of *SARM1* alleles encoding ARM domain and other N-terminal variants aggregated from all datasets used in this study.

| | Variant | Study alleles* | | | Study MAF* | | |
| | | Cases | Controls | gnomAD alleles† | Cases | Controls | gnomAD MAF† |
|---|---|---|---|---|---|---|---|
| | | /22,234 | /20,804 | /63,882 | | | |
| Strong GoF | L223P | 1 | 0 | 0 | 0.000045 | 0 | 0 |
| | Δ229–235 | 5 | 0 | 4 | 0.00022 | 0 | 0.000063 |
| | Δ249–252 | 1 | 0 | 0 | 0.000045 | 0 | 0 |
| | R267W | 2 | 0 | 1 | 0.000090 | 0 | 0.000016 |
| | V331E | 1 | 0 | 0 | 0.000045 | 0 | 0 |
| | E340K | 1 | 0 | 4 | 0.000045 | 0 | 0.000063 |
| | T385A | 2 | 0 | 0 | 0.000090 | 0 | 0 |
| | Combined | 13 | 0 | 9 | 0.00058 | 0 | 0.00014 |
| Other | G35E | 1 | 0 | 1 | 0.000045 | 0 | 0.000016 |
| | T39M | 0 | 1 | 0 | 0 | 0.000048 | 0 |
| | L76fs | 0 | 1 | 0 | 0 | 0.000048 | 0 |
| | V88M | 0 | 1 | 0 | 0 | 0.000048 | 0 |
| | V112I | 1 | 0 | 1 | 0.000045 | 0 | 0.000016 |
| | R131G | 1 | 0 | 1 | 0.000045 | 0 | 0.000016 |
| | A210V | 0 | 1 | 0 | 0 | 0.000048 | 0 |
| | A240E | 1 | 0 | 0 | 0.000045 | 0 | 0 |
| | R244S | 1 | 0 | 0 | 0.000045 | 0 | 0 |
| | A250T | 1 | 0 | 0 | 0.000045 | 0 | 0 |
| | A275V | 1 | 2 | 2 | 0.000045 | 0.000096 | 0.000031 |
| | A301S | 6 | 1 | 8 | 0.00027 | 0.000048 | 0.00013 |
| | R310H | 2 | 2 | 11 | 0.000090 | 0.000096 | 0.00017 |
| | N337D | 3 | 4 | 26 | 0.00013 | 0.00019 | 0.00041 |
| | A341V | 3 | 2 | 8 | 0.00013 | 0.000096 | 0.00013 |
| | Combined | 21 | 15 | 58 | 0.00094 | 0.00072 | 0.00091 |
| | P332Q* | 198 | 206 | 789 | 0.0089 | 0.010 | 0.012 |

*Aggregated values for all datasets listed in Figure 8.

†Values for the non-neuro European (non-Finnish) population in gnomAD v3.1.1. These values do not include case samples collected as part of a neurologic or psychiatric case/control study. However, that does not guarantee that any carriers of the alleles listed are not already affected or will not go on to develop a motor nerve disorder.

duplication among the group, this reveals a broad range of clinical presentation and a highly variable age-of-onset among the cases, with no obvious heritability.

The lack of any clear and consistent trend in disease severity probably indicates that SARM1 GoF is not the only driver of disease in some or all of these cases. In this respect it is notable that two of the ALS cases are also heterozygous for I114T (previously reported as I113T) *SOD1* and L17P *FIG4* variant alleles that have previously been linked to ALS or recessive Charcot-Marie-Tooth disease type 4 J (CMT4J). Given that the I114T *SOD1* allele shows incomplete penetrance (*Lopate et al., 2010*) and the L17P *FIG4* allele has previously only been associated with CMT4J in combination with a null allele (*Nicholson et al., 2011*), it raises the possibility that an interaction with the *SARM1* GoF alleles is critical for pathogenesis in these ALS cases. Therefore, an interaction between *SARM1* GoF alleles and other risk alleles may enhance phenotypic expressivity and underlie some of the variability in clinical presentation.

**Table 4.** Clinical information for motor nerve disorder patients heterozygous for *SARM1* GoF alleles.

| Clinical diagnosis (site of onset) | SARM1 variant | Cohort | Sex | Age-of-onset / progression | Other variants | Close relatives affected | Other observations / co-morbidities |
|---|---|---|---|---|---|---|---|
| ALS (bulbar) | L223P | Project MinE | M | 59 / 3 yr. dead | None * | Not known | |
| ALS (bulbar) | Δ229–235 | Answer ALS | F | 70 / 2 yr. dead | None† | Not known | - No FTD<br>- *Hyperlipidaemia* |
| ALS | Δ229–235 | GENESIS | M | 66 / 1 yr. dead | ‡SOD1 | None | - Rapid progression |
| ALS (spinal) | Δ229–235 | Unattributed | F | 71 / 2 yr. dead | Not available | None | - Lower extremity onset<br>- *FTD* |
| ALS (spinal) | Δ229–235 | Project MinE | M | 42 / 3 yr. dead | None * | Not known | |
| ALS (spinal) | Δ229–235 | Project MinE | M | 57 / 3 yr. dead | None * | Not known | |
| ALS (bulbar) | Δ249–252 | Project MinE | M | 65 / 4 yr. dead | None * | Not known | |
| ALS (spinal) | R267W | Answer ALS | M | 58 / 2 yr. dead | §FIG4† | Not known | - Lower limb onset<br>- *COPD* and *Type II diabetes* |
| ALS (spinal) | V331E | Project MinE | M | 53 / 5 yr. dead | None * | Not known | |
| ALS (spinal) | E340K | Project MinE | M | 75 / 6 yr. dead | None * | Not known | |
| ALS (spinal) | T385A | Project MinE | M | 69 / 6 yr. alive | None * | Not known | |
| ALS (bulbar) | T385A | Project MinE | F | 63 / 1 yr. dead | None * | Not known | |
| HSP (pure) | Δ229–235 | GENESIS | F | 5 / 14 yr. alive | Not available | None | - Presented with lower extremity spasticity and hyperreflexia extensor plantar responses<br>- Wheelchair use at 19 yr.<br>- Normal total spine MRI (smallish thoracic cord) |
| Upper and lower motor nerve disorder | R267W | UCL | M | 40 / 25 yr. alive | None ¶ | None | - Presented with right leg weakness, thigh fasciculations and slowly progressive wasting<br>- No sensory, bladder, bowel, or bulbar involvement<br>- Mild hand weakness and brisk reflexes by 50 yr.<br>- Lower limb spasticity by 61 yr.<br>- Still ambulates independently<br>- Proximal lower limb denervation> distal upper limb denervation. |
| ALS (spinal) | V112I | Project MinE | M | 52 / 17 yr. dead | None * | Not known | |

FTD; frontotemporal dementia, COPD; chronic obstructive pulmonary disease.

*Screened for variation in *C9orf72*, *FUS*, *SOD1* and *TARDBP*.

†Screened for variation in *ALS2*, *ATXN2*, *C9orf72*, *CCNF*, *CHCHD10*, *FUS*, *OPTN*, *PFN1*, *SOD1*, *TARDBP*, *TBK1*, *UBQLN2*, *VAPB* and *VCP*.

‡Heterozygous for the partially-penetrant I114T *SOD1* variant associated with sporadic ALS.

§Heterozygous for the L17P Figure 4 variant, a variant that has previously only been associated with CMT4J in combination with a null allele.

¶Screened for variation in *Androgen Receptor* expansion, *ALS2*, *ANG*, *C9orf72*, *CYP7B1*, *DCTN1*, *FA2H*, Figure 4, *FUS*, *GJC2*, *KIAA0196*, *KIF5A*, *NIPA1*, *OPTN*, *PLP1*, *PNPLA6*, *REEP1*, *RTN2*, *SPAST*, *SPG7*, *SLC52A1*, *SLC52A2*, *SLC52A3*, *SMN*, *SOD1*, *TARDBP*, *UBQLN2*, *VAPB*, *VCP*, whole mtDNA sequencing in blood.

## Discussion

Here, we have identified seven disease-associated *SARM1* alleles in patients with ALS or other motor nerve disorders that encode missense or in-frame microdeletion variants with constitutive NADase activity as much as ten to twenty times higher than that of WT SARM1. This level of activity is comparable to that of SARM1 lacking any auto-inhibition and is substantially higher than the activity of WT SARM1 in the presence of NMN, a natural activator of SARM1 NADase during injury-induced and other related forms of axon degeneration. Although co-expression experiments suggest that the NADase activity of hetero-oligomers of WT and strong NADase GoF SARM1 will be around half that of homo-oligomers of strong GoF SARM1 alone, this still represents a considerable increase in activity. Consistent with this, acute expression of strong NADase GoF SARM1 alongside WT SARM1 in primary cultured neurons is sufficient to promote robust NADase-dependent cell death (and simultaneous or secondary axon degeneration) in response to otherwise non-harmful mild stress. Crucially, these seven strong GoF alleles are enriched in the motor nerve disorder cohorts studied compared to control groups or the wider population implicating constitutively hyperactivated SARM1 as a risk factor for these diseases.

Despite the robust association between SARM1 strong GoF variants and disease in these cohorts, the proportion of motor nerve disorder cases that are heterozygous for the strong GoF alleles identified here is only 0.12 %. However, other ALS patient-associated *SARM1* alleles encoding variants with predicted strong NADase GoF have been identified concurrently with this study (*Bloom et al., 2021*), and there are many other ARM domain variant alleles among our extended datasets that remain untested and could, therefore, include additional GoF alleles. The real percentage may thus be higher. Furthermore, in light of this association, we expect that other SARM1 GoF and/or activation mechanisms will also be found to play a key role in motor nerve disorders. For example, the GWAS association between a *SARM1* intragenic SNP and ALS (*Fogh et al., 2014*; *van Rheenen et al., 2016*) could be driven by increased SARM1 expression as a result of more common variants which impact transcriptional regulation of the *SARM1* gene. Reduced expression of Stathmin-2 (STMN2, also known as SCG10), which potentiates programmed axon degeneration (*Shin et al., 2012*), has also been linked to the disease via a mechanism that is expected to involve aberrant SARM1 activation. Specifically, STMN2 depletion increases the vulnerability of ALS patient iPSC-derived neurons (*Klim et al., 2019*; *Melamed et al., 2019*), and a non-coding *STMN2* variant that influences gene expression has recently also been associated with increased risk of sporadic disease (*Theunissen et al., 2021*). An age-related decline in NMNAT2 expression and/or delivery to axons or exposure to environmental toxins could also result in aberrant SARM1 activation and increased risk of disease (*Milde et al., 2015*; *Loreto et al., 2021*; *Wu et al., 2021*).

The association between motor nerve disorders and functionally confirmed strong GoF variants of SARM1 we report here is highly significant at the individual gene level and contrasts with the absence of disease association for neighbouring variants with properties more comparable to WT SARM1 (thereby representing good controls for testing our hypothesis). Thus, unlike a conventional gene burden analysis, it is the functional consequence of each variant, rather than its location, that correlates with disease. The low allele frequency of the GoF variants, even after their enrichment in ALS, means there is insufficient power to show genome-wide significance, although the lead SNP in an ALS GWAS study, rs35714695, which is far more common and hence does show genome-wide significance, is located only 8.3–11.4 kb from each of these strong GoF variants within the same ~200 kb linkage disequilibrium block (*Fogh et al., 2014*; *van Rheenen et al., 2016*). Although these are essentially independent observations, as it is not feasible to confirm linkage disequilibrium between such rare SNPs and rs35714695, they separately support the notion that SARM1 contributes to ALS. Hypothesis-driven studies such as this one are, of course, not unbiased in the way that GWAS is, but instead depend substantially on the strength of the hypothesis and the availability of clear functional data to support it. However, where these are both provided they have a vital role to play in complementing GWAS studies by closing the large heritability gap in ALS and other disorders (*Van Damme, 2018*), since genome wide significance for genuinely pathogenic, but very rare alleles, may not be an achievable goal. The current study also illustrates the importance of publicly available datasets such as Project MinE in making such work possible.

Among the SARM1 variants tested for function in this study, the seven with strong NADase GoF can be clearly differentiated from the rest. However, we cannot rule out that a more modest increase in constitutive NADase GoF, directly demonstrated for some variants in NADase assays of recombinant protein (V88M, V112I, and A210V SARM1), or possibly implied for others based on their ability to modestly enhance NAD$^+$ loss in HEK cells (A275V and A341V SARM1), might also have biologically-relevant consequences. It was also surprising to us that raised constitutive activity was the only GoF effect we identified among the ALS patient-associated *SARM1* alleles, with no evidence of enhanced triggering by NMN from a normal basal level. While our recombinant protein assays provide a useful insight into the relative NMN-responsiveness of the different variants, and can clearly identify reduced NMN sensitivity, it is possible that some GoF relating to enhanced inducibility may not yet have been identified and that this could contribute to pathogenesis in some patients.

Recently, substantial progress has been made into elucidating the mechanism of auto-inhibition of SARM1 NADase activity by the ARM domain (*Jiang et al., 2020*; *Sporny et al., 2020*; *Shen et al., 2021*) and the strong GoF variants identified here could represent useful tools for further understanding. Given their vastly higher activities relative to NMN-activated WT SARM1, it seems unlikely that the amino acid changes in the strong GoF variants simply mimic the effects of NMN binding and/or block NAD$^+$ binding in the allosteric site in the ARM domain and, notably, there is little direct overlap

with residues that appear to be critical for NAD$^+$/NMN interactions (*Figley et al., 2021*). Instead, constitutive activities similar to the activity of the SAM-TIR fragment of SARM1 lacking the entire auto-inhibitory ARM domain suggest that the amino acid changes probably cause more profound structural changes that disrupt inhibitory interaction between the ARM and TIR domains completely. This would also explain the complete unresponsiveness of the variants to NMN. Consistent with this, some of the changes in the strong GoF variants involve residues located at an ARM-TIR interface (R249 and E252), an ARM-SAM interface (V331) or an ARM-ARM interface (T385) (*Shen et al., 2021*).

Given the potent pro-death effect of constitutively hyperactivated SARM1 in primary neuronal cultures, the heterogeneous clinical presentation of motor nerve disorder patients with strong GoF *SARM1* alleles is surprising. In fact, that they survive at all with such high constitutive SARM1 NADase activity is, in itself, unexpected given that our data indicate that WT SARM1 in the presence of NMN is less active than hetero-oligomers of WT and strong GoF SARM1 but is still predicted to be responsible for the early lethality in both mice and humans lacking functional NMNAT2 (*Gilley et al., 2013*; *Gilley et al., 2015*; *Lukacs et al., 2019*). We therefore propose that chronic, physiological expression of constitutively hyperactivated SARM1 specifically triggers compensatory responses during development to limit its impact. Suppressed expression of SARM1 itself, as seen in transfected HEK 293T cells, is one possible compensatory change that could restrict NADase activity *in vivo*, although this seems to be a direct consequence of acute and extensive SARM1-dependent depletion of NAD$^+$ (and/or ATP) resulting in suppression of *de novo* protein synthesis and it is hard to imagine long-term neuron survival under these conditions. Intriguingly, differential responses to constitutive or acute silencing of a single *Nmnat2* allele is also suggestive of compensatory changes in the chronic condition (*Gilley et al., 2013*) and it is tempting to speculate that they might be related.

It will also be important to firmly establish the extent of the penetrance of the GoF alleles identified here and whether any show a familial association with motor nerve disorders. However, their presence in sporadic cases, and co-occurrence with partially-penetrant disease variants of other genes in some patients, suggest that they have to interact with other risk factors to reveal their pathogenicity. Future human genetics studies should help to address these possible associations. There are also a number of drug development programmes and other avenues of therapeutic intervention targeting SARM1 currently being pursued (*Geisler et al., 2019*; *Hughes et al., 2021*; *Bosanac et al., 2021*) and individuals with *SARM1* GoF alleles would thus be ideal test subjects if other requirements are met.

In summary, we report seven rare *SARM1* alleles encoding variants with constitutive NADase activity much higher than NMN-activated wild-type SARM1, and broadly comparable to that of completely uninhibited SARM1, that show clear association with ALS and other motor nerve disorders in a small percentage of patients. There is phenotypic heterogeneity in the clinical presentation, reminiscent of other ALS disease genes, and this raises intriguing questions both about the potential pathogenic role of constitutively hyperactive SARM1 and about why their extraordinarily high NADase activity is not immediately lethal. Association of function-altering coding variants also supports the notion that SARM1 has a wider contribution to ALS. While further replication in other cohorts is important, along with direct evidence that SARM1 activity is necessary for pathogenesis to firmly establish causation, the results presented here strongly implicate SARM1 as a risk factor in this group of diseases.

## Materials and methods
### Constructs and reagents

The bicistronic pLVX-IRES-ZsGreen1 vector (Clontech / Takara Bio 632187) was the backbone for all SARM1 expression constructs used in this study, unless stated otherwise. SARM1 and ZsGreen(1) are co-expressed from a bicistronic mRNA transcript under the control of the CMV promoter. The complete open reading frame (ORF) of the canonical 724 amino acid human SARM1 protein was cloned into the first cistron between the *Eco*RI and *Spe*I restriction sites. An in-frame Flag tag followed by a stop codon were introduced between the *Xba*I and *Bam*HI restriction sites immediately 3' to SARM1 ORF at the C-terminus. The expressed SARM1 thus possesses a C-terminal Flag (after a four amino acid linker). ZsGreen occupies the second cistron and undergoes IRES-driven translation. For experiments in *Figure 6—figure supplement 1*, untagged SARM1 was expressed from a pCMV-Tag4A vector backbone (Agilent Technologies) with the 724 amnio acid ORF, including a consensus Kozak sequence immediately 5' to the ATG codon and a stop codon at the 3' end of the ORF, cloned

between the *Bam*HI and *Xho*I restriction sites of the multiple cloning site. Point mutations were introduced into constructs by QuikChange II site-directed mutagenesis (Stratagene) and microdeletions were introduced using overlap extension PCR (*Lee et al., 2010*). The presence of the mutations and absence of other PCR errors was confirmed by DNA sequencing (Department of Biochemistry, University of Cambridge). pDsRed2-N1 (Clontech) was used for expression of variant *Discosoma* red fluorescent protein (DsRed) to label microinjected neurons and their neurites.

PBS without calcium and magnesium (Merck) was used throughout, unless stated otherwise. $NAD^+$ and NMN (both Merck) were dissolved in water and stored frozen as concentrated stocks. Nicotinamide riboside (NR) was prepared as a 100 mM stock from Tru Niagen capsules by dissolving the contents in PBS and passing through a 0.22 μm filter. NAMPT inhibitor FK866 was a kind gift from Prof. Armando Genazzani, University of Novara.

## Cell lines and transfection

Human embryonic kidney (HEK) 293T cells (clone 17, [HEK 293T/17]) were obtained from the American Type Culture Collection (ATCC, CRL-11268, RRID:CVCL_1926) and had been authenticated by STR profiling. Mycoplasma contamination was not detected. Cells were grown in DMEM with 4500 mg/L glucose and 110 mg/L sodium pyruvate (PAA), supplemented with 2 mM glutamine, 1 % penicillin/streptomycin (both Invitrogen), and 10 % fetal bovine serum (PAA). Cells at 50–70% confluence were transfected with expression constructs (as described in the text / figure legends) using Lipofectamine 2000 reagent (Invitrogen) according to the manufacturer's instructions. Amounts of DNA constructs transfected are described in the figure legends.

## Measurements of nucleotide levels in transfected HEK 293T cell extracts

Transfected HEK 293T cells (from individual wells of 24-well plates) were collected and washed in ice-cold PBS 24 hr after transfection and lysed in ice-cold KHM buffer (110 mM potassium acetate, 20 mM HEPES pH 7.4, 2 mM $MgCl_2$, 0.1 mM digitonin) containing cOmplete, Mini, EDTA-free protease inhibitor cocktail (Roche). Efficient lysis with KHM buffer required an initial trituration by pipetting and intermittent vortexing over a 10 min incubation on ice. Lysates were centrifuged for 5 min at 3000 rpm in a microfuge at 4 °C to pellet insoluble material, and supernatants were then collected on ice and diluted to 0.5 μg/μl in cold KHM buffer after protein concentrations had been determined using the Pierce BCA assay (Thermo Fisher Scientific). The NAD/NADH-Glo and NADP/NADPH-Glo assays (Promega) were used to measure $NAD^+$ and $NADP^+$ in extracts. First, 25 μl of each extract was mixed with 12.5 μl 0.4 M HCl and heated to 60 °C for 15 min to degrade NADH and NADPH before being allowed to cool to room temperature (RT) for 10 min. Reactions were then neutralised by adding 12.5 μl 0.5 M Tris base and 10 μl of each neutralised reaction was mixed with 10 μl of the relevant detection reagent on ice in wells of a 384-well white polystyrene microplate (Corning). Once all reactions had been set up the plate was moved to a GloMax Explorer plate reader (Promega) and incubated for 40 min at 25 °C before reading for luminescence. The CellTiter-Glo assay (Promega) was used to measure ATP in extracts. Here, 10 μl of extract was mixed directly with 10 μl of assay reagent on ice in wells of a 384-well white polystyrene microplate. Luminescence was then read after 10 min at 25 °C using a GloMax Explorer plate reader. Concentrations of nucleotides were determined from standard curves generated from dilution series of the relevant nucleotides and then calculated as proportions of the amount in extracts from cells transfected in parallel with empty vector.

## Immunoprecipitation of recombinant SARM1 from HEK 293T cells

Recombinant, C-terminal Flag-tagged or HA-tagged human SARM1 (WT or variant) was purified from HEK 293T cells by immunoprecipitation. HEK 293T cells in 10 cm and 6 cm dishes and individual wells of a 6 cm dish were respectively transfected with 24 μg, 10 μg and 4 μg of SARM1 expression construct. Media was supplemented with 2 mM NR at the time of transfection. Cells were lysed in KHM buffer as for the measurements of nucleotide levels in transfected HEK 293T cells (see above) but extracts were instead diluted to 1 μg/μl in cold KHM buffer after determining protein concentration. For immunoprecipitation, extracts were incubated overnight at 4 °C with rotation with 20 μg/ml mouse ANTI-FLAG M2 monoclonal antibody (Sigma-Aldrich F3165, RRID:AB_259529) or 2 μg/ml mouse anti-HA monoclonal antibody, clone 12CA5 (Sigma-Aldrich 11583816001, RRID:AB_514505)

and 50 μl/ml of pre-washed Pierce magnetic protein A/G beads (Thermo Fisher Scientific). For co-immunoprecipitations, beads were collected on a magnetic rack and washed 3 x with KHM buffer and 1 x with PBS (with protease inhibitors) and then resuspended in PBS containing 1 μg/μl BSA. Beads were resuspended in 150 μl per ml of the original immunoprecipitation volume for normal yield proteins, or 25–50 μl for low yield proteins. Inclusion of BSA in the re-suspension buffer allowed samples to be frozen without substantial loss of SARM1 activity (activity is almost completely lost after one freeze/thaw cycle without its inclusion). The concentrations of SARM1 proteins in the bead suspension were determined by comparison to a standard curve generated from a dilution series of recombinant SARM1 of known concentration (purified from insect cells) after detection by immunoblotting using a rabbit polyclonal antibody raised against the SAM domains of human SARM1 (both kindly provided by AstraZeneca).

Of particular note, the use of KHM lysis buffer is critical for subsequent assaying of the NADase activity of the purified SARM1 protein. Harsher lysis buffers with high concentrations of strong detergents (e.g. RIPA-based buffers) themselves cause substantial activation of SARM1 NADase and a loss of responsiveness to NMN.

## NADase assays of recombinant purified SARM1

A series of test assays were first performed to define appropriate concentrations of immunoprecipitated SARM1 proteins and the earliest, reliable assay timepoints to obtain rates as close as possible to the maximal rate under the various reaction conditions used. Optimised reaction conditions were as follows: 25 μl reactions (overall 1 x PBS) contained 1 ng/μl SARM1 protein for strong GoF variants (L223P, Δ229–235, Δ249–252, V331E, E340K and T385A) or SAM-TIR fragment, or 5 ng/μl of WT SARM1 and other variants, together with 25 μM $NAD^+$ ± 10 μM or 50 μM NMN. Reactions were kept on ice until all had been set up. All reactions were performed with the recombinant SARM1 still attached to beads as inefficient elution severely restricted the yield of the poorly expressed GoF variants. Bead suspensions were thoroughly mixed prior to addition to the reactions. Constitutive (basal) $NAD^+$ consumption was measured from reactions containing just $NAD^+$ as the difference between starting levels (0 hr) and levels remaining after a 1 hr (for strong GoF variants) or 2 hr (for all other variants) incubation at 25 °C. $NAD^+$ consumption in the presence of NMN (both 10 μM and 50 μM) was calculated after a 1 hr incubation for all variants. Reactions were each mixed once mid-way through the incubation period to resuspend the beads. To measure $NAD^+$ using the NAD/NADH-Glo assay, 5 μl aliquots of the reaction were removed immediately after setting up (whilst still on ice), to obtain precise starting levels in individual reactions, and after the defined times listed above. Aliquots were then mixed with 2.5 μl of 0.4 M HCl, to stop the reaction, and neutralised by mixing with 2.5 μl 0.5 M Tris base after 10 min. To bring $NAD^+$ concentrations down to the linear range of the NAD/NADH-Gloassay, neutralised samples were subsequently diluted 1 in 50 in a buffer consisting of 50 % PBS, 25 % 0.4 M HCL, 25 % 0.5 M Tris base. Ten μl of the diluted sample was then mixed with 10 μl of detection reagent on ice in wells of a 384-well white polystyrene microplate (Corning). Once all reactions had been set up the plate was moved to a GloMax Explorer plate reader (Promega) and incubated for 40 min at 25 °C before reading for luminescence. $NAD^+$ concentrations were determined from standard curves (as above). $NAD^+$ consumption rates were converted to mol consumed per min per mol of SARM1 protein (mol/min/mol SARM1). This allows a direct comparison of rates between full-length SARM1 and the SAM-TIR fragment. Individual data points for each separate protein preparation are the means of two technical replicates in most cases. Non-specific activity on bead/antibody complexes in control immunoprecipitations (from HEK 293T cells transfected with empty vector) was negligible at less than 0.5 % of the constitutive $NAD^+$ consumption seen for WT SARM1 (average of n = 4 assays).

## Immunoblotting

Protein extracts from transfected HEK 293T cells used for nucleotide measurements (0.5 μg protein/μl, see above) were diluted 1 in 2 with 2 x SDS-PAGE loading buffer, and bead suspensions of immunoprecipitated recombinant SARM1 proteins were diluted 1 in 16 for immunoblotting. Samples were incubated at 100 °C for 3 min and 10 μl of sample per lane resolved on 4–20% gradient gels (Bio-Rad) before being transferred to Immobilon-P membrane (Millipore). Blots were blocked in TBS (20 mM Tris p.H. 8.3, 150 mM NaCl) containing 5 % skim milk powder for 30 mins at RT before being incubated overnight at 4 °C with primary antibody in TBS containing 0.05 % Tween-20 (Merck) (TBST) and 5 % milk.

After 3 × 10 min washes in TBST, blots were incubated for 1–2 hr at RT with appropriate HRP-conjugated secondary antibodies (diluted 1 in 3,000, Bio-Rad) in TBST with 5 % milk. After 3 × 10 min washes in TBST and one rinse in TBS, blots were incubated with Pierce ECL Western Blotting Substrate or SuperSignal West Dura Extended Duration Substrate (both Thermo Fisher Scientific) and imaged using an Alliance chemiluminescence imaging system (UVITEC Cambridge). Relative band intensities on captured digital images were determined from areas under histogram peaks using Fiji software (https://fiji.sc).

The following primary antibodies were used: mouse ANTI-FLAG M2 monoclonal antibody (Sigma-Aldrich F3165, RRID:AB_259529, 2 µg/ml as standard or 0.5 µg/ml for probing immunoblots of co-immunoprecipitations), mouse anti-ZsGreen1 monoclonal antibody, clone OTI2C2 (OriGene TA180002, RRID:AB_2622267, 1 in 1000), mouse anti-GAPDH monoclonal antibody, clone 6C5 (Abcam ab8245, RRID:AB_2107448, 1 in 2000), mouse anti-HA monoclonal antibody, clone 12CA5 (Sigma-Aldrich 11583816001, RRID:AB_514505, 0.2 µg/ml as standard or 0.05 µg/ml for immunoblots of co-immunoprecipitations), a rabbit polyclonal antibody raised against the SAM domains of human SARM1 (kindly provided by AstraZeneca, 1 in 2000) and a mouse anti-SARM1 monoclonal antibody (*Chen et al., 2011*) (1 in 2,000).

## Primary SCG neuron cultures and microinjection

SCG neuron cultures were prepared from wild-type P1-P3 mouse pups (C57BL/6 J, RRID:IMSR_JAX:000664), or *Sarm1*$^{-/-}$ P1-P3 pups (RRID:MGI:3765957) for *Figure 6—figure supplement 1* only, and were microinjected following established protocols (*Gilley and Loreto, 2020*). Mice were used in accordance with the Home Office Animal Scientific Procedures Act (ASPA), 1986 under project licence P98A03BF9. Dissected SCG ganglia were incubated at 37 °C for 20 min in 0.025 % trypsin (Sigma) dissolved in PBS followed by 20 min in 0.2 % collagenase type II (Gibco) in PBS (with calcium and magnesium). After, ganglia were gently triturated using a pipette tip in SCG culture medium comprising Dulbecco's Modified Eagle's Medium (DMEM) (4500 mg/L glucose and 110 mg/L sodium pyruvate, Sigma) supplemented with 2 mM glutamine (Invitrogen), 1 % penicillin/streptomycin (Invitrogen), 10 % fetal bovine serum (Sigma), 2 µg/ml aphidicolin (Calbiochem), and 100 ng/ml 7 S NGF (Invitrogen). After a 1–2 hr pre-plating stage to remove non-neuronal cells, dissociated neurons were plated in a small laminin-coated area on ibidi µ-dishes pre-coated with poly-L-lysine (Thistle Scientific). Cultures were maintained in SCG medium, which was changed every 2–3 days, and were injected after 5–7 days. The inclusion of aphidicolin in the medium suppressed growth and proliferation of any non-neuronal cells.

SCG neurons were microinjected using a Zeiss Axiovert 200 microscope with an Eppendorf 5,171 transjector and 5,246 micromanipulator system and Eppendorf Femtotips. Expression constructs were diluted in 0.5 x PBS and cleared by spinning through a Spin-X filter (Costar). The mix was injected into the nuclei of SCG neurons. The concentrations of expression constructs in the injection mix are listed in figure legends. A total of 40–45 neurons were injected per dish. Although ZsGreen is co-expressed from the SARM1 vector used in these studies, restricting the amount of vector injected to limit SARM1 expression meant that the resulting ZsGreen expression was itself not sufficient for clear visualisation of injected neurons and their neurites. For the purposes of visualisation we thus had to co-inject a DsRed expression construct at a higher concentration.

## Imaging

Fluorescence and phase contrast images of transfected HEK 293T cells and injected SCG neurons were acquired with a Leica DFC365FX fluorescence monochrome camera attached to a Leica DMi8 inverted fluorescence microscope using 10 x and 20 x objectives.

## Database resources

The different datasets used in this study are annotated to different genome builds, either Genome Reference Consortium Human Build 37 (GRCh37), also known as hg19 (for Project MinE, GENESIS and the HSP cohort) or GRCh38/ hg38 (for Answer ALS, UCL and the LBC). GRCh37 is annotated to a non-canonical *SARM1* cDNA and there are sequence irregularities compared to GRCh38 (see *Figure 1* legend). Therefore, throughout this study we only report *SARM1* genetic variation using numbering for genome build to GRCh38 and the canonical 724 amino acid SARM1 protein (having converted all GRCh37-based information from datasets that use that build). For this reason, we also use population MAFs reported in gnomAD v3.1.1 in this study as it is referenced to GRCh38 (as opposed to gnomAD

v2.1.1 which is referenced to GRCh37 and consequently contains a number of MAF discrepancies and additionally a lack of coverage for some alleles).

Many neurological disorders are represented in the complete GENESIS and UCL databases. Therefore, for the purposes of this study, our disease cohorts are restricted to individuals with motor nerve disorders and our control cohorts are restricted to unaffected individuals and individuals with non-neurological conditions. For GENESIS, the disease cohort consisted of individuals with ALS, pure or complex HSP, distal hereditary motor neuropathy and frontotemporal dementia with motor neuron disease, and the control cohort consisted of unaffected individuals, individuals with dilated cardiomyopathy and small numbers of individuals with metabolic or congenital disorders. For the UCL dataset, the disease cohort consisted of cases of spinal muscular atrophy and related disorders (including ALS), HSP and hereditary motor and sensory neuropathy, and the control cohort consisted of unaffected (control) individuals and a smaller number of individuals with Cluster headache syndrome.

Sample sequencing, variant calling and quality control methods for each dataset are broadly comparable. These are described elsewhere for Project MinE (*Project MinE ALS Sequencing Consortium, 2018*) and the LBC (*Wragg et al., 2020*). Information for *SARM1* variants in the Answer ALS project was obtained using ANNOVAR (*Wang et al., 2010*) annotation on GRCh38 positions, after read mapping with Burrows-Wheeler Alignment tool (BWA) (*Li and Durbin, 2010*), variant calling with GATK (*McKenna et al., 2010*), and joint-genotype using Sentieon (*Freed et al., 2017*). For GENESIS samples, Illumina WGS sequencing samples were aligned to the GRCh37 reference using BWA. Duplicate reads were marked using Picard's MarkDuplicates utility. The bam file containing de-dupped reads was then used to call single nucleotide variations and indels using Freebayes, prior to October 2018, or later with GATK (v4). For the UCL rare disease dataset, exome sequencing was performed using the Agilent SureSelect Human All Exon V6 enrichment and enriched libraries underwent 150 base pair, paired-end sequencing on an Illumina HiSeqX10 next-generation sequencing platform. Sequence data were aligned to GRCh38 and variant calling using GATK with filtering of PCR duplicates. For the HSP study samples, WES and WGS sequencing data were processed by GSvar pipeline (https://github.com/imgag/ngs-bits; *Schroeder et al., 2017*) and sequencing files were aligned to GRCh37 human reference by BWA. Variants were called by GATK v4 with filtering of PCR duplicates. It was not feasible within the context of this study to validate variants by obtaining raw sequencing data or DNAs for Sanger sequencing but there is strong confidence in calling all variants tested in this study based on phred-scaled quality scores, when available.

Although complete ancestry information is not available to us, the vast majority of samples in all datasets used in this study, other than those in the GENESIS cohorts, are expected to be of predominantly European ancestry, and even within the GENESIS cohorts a smaller majority are expected to be of European origin. This is broadly supported by the observed frequency of the P332Q *SARM1* allele in each cohort, an allele that is relatively more common in the European (non-Finnish) population than other populations (*Figure 8*). As such, ancestry is not expected to have a substantial effect on the association statistics we have performed using aggregated data.

No new datasets or code were generated for this study. The Project MinE initiative data browser for Data Freeze one for searching genetic variation in patient and control cohorts was accessed at http://data browserprojectmine.com/. Access to additional WGS / WES data in Project MinE (for DF2) and the other databases explored in this study was obtained upon request. Researchers can apply for similar access via the following links or lead investigators (who can be contacted via the corresponding author):

Project MinE - https://www.projectmine.com/research/data-sharing/
Answer ALS - https://www.nygenome.org/als-consortium/
GENESIS - https://neuropathycommons.org/genetics/genesis-platform
UCL rare disease (neurology) dataset - Prof. Henry Houlden (listed author).
HSP study - Dr. Rebecca Schüle (listed author).
Lothian Birth Cohort - https://www.ed.ac.uk/lothian-birth-cohorts/data-access-collaboration

There may be restrictions for commercial companies in some cases.

## Patient details

This is a retrospective study using anonymised data so specific consent was not obtained by the authors, but consent was obtained at each site that contributed patient information to this study in accordance with their local Institutional Review Boards (IRBs). Patient information for the Answer ALS

project patients (patient IDs NEUMA411HJP and NEUBD218YR3) was already publicly available at https://dataportalanswerals.org/search.

## Statistics

Multiple comparisons, multiple paired *t* tests and one- or two-way ANOVA were performed using Prism software (GraphPad Software Inc, La Jolla, CA, USA) as described in the figure legends. Log transformation of the data was performed prior to statistical testing where appropriate as indicated in figure legends and source data files; however, non-transformed data were plotted on graphs in all Figures for ease of interpretation. A p value of <0.05 was considered significant and values above this cut-off are considered not significant (ns). For tests applying a false discovery rate (FDR) correction, the adjusted p value we report is the q value.

To determine whether there is a burden of rare variants (minor allele frequency <0.01) in the *SARM1* gene based on their functional properties (as determined in this study), we performed burden and optimized sequence kernel association test (SKAT-O) as implemented in the SKAT-O R package (*Lee et al., 2012*). 'Burden' refers to a linear sum of $\chi^2$ random variables where each variable is assigned a genetic variant (*Davies, 1980*).

## Acknowledgements

This work was supported by a grant from the Robert Packard Center for ALS Research at Johns Hopkins (to MPC, also supporting JG) and in part by data provided by the Answer ALS Consortium (JK and LL) – also administered by the Robert Packard Center. Its contents are solely the responsibility of the authors and do not necessarily represent the official views of The Johns Hopkins University or any grantor proving funds to its Robert Packard Center for ALS Research. Parts of this work were also supported by UK Biotechnology and Biological Sciences Research Council (BBSRC) / AstraZeneca Industrial Partnership award BB/S009582/1 (to MPC, also supporting JG, OJ) and Wellcome Trust Collaborative award 220906/Z/20/Z (to MPC and MMR, also supporting JG, OJ, MP). Additional support was provided by UK Medical Research Council (MRC) awards MR/L501529/1 and MR/R024804/1 and the UK Economic and Social Research Council (ESRC) award ES/L008238/1 under the aegis of the EU Joint Programme - Neurodegenerative Disease Research (JPND) (http://www.jpnd.eu) (to AA-C), the Motor Neurone Disease (MND) Association and the National Institute for Health Research (NIHR) Biomedical Research Centre at South London and Maudsley NHS Foundation Trust and King's College London (AA-C, AI), National Institute of Neurological Diseases and Stroke (NINDS) and office of Rare Diseases awards U54NS065712 (to MMR, also supporting MP), and 5R01NS072248-10 and 5R01NS105755-03 (to SZ, also supporting MCD), National Institute of Environmental Health Sciences (NIEHS) award K23ES027221 (to SAG), and Wellcome Trust award 216596/Z/19/Z (to JCK). Samples used in this research were in part obtained from the UK National DNA Bank for MND Research, funded by the MND Association and the Wellcome Trust. Sample management was undertaken by Biobanking Solutions funded by the MRC at the Centre for Integrated Genomic Medical Research, University of Manchester. Whole-genome sequencing of the Lothian Birth Cohorts was funded by the Biotechnology and Biological Sciences Research Council. Anne Segonds-Pichon kindly helped with statistical analyses. We are grateful to the following for accessing Project MinE data: Evelijn Zeijdner and Jan Veldink (UMC Utrecht). We also thank the following for useful discussion and feedback: Jemeen Sreedharan, Ahmet Höke, David Bennett, Steve Finkbeiner, Giuseppe Orsomando and all members of the Coleman group. We would like to thank people with MND/ALS and their families for their participation in this project.

## Additional information

### Competing interests
Jonathan Gilley: Part-funded by AstraZeneca during the past 3 years. Oscar Jackson: Currently part-funded by AstraZeneca. Michael P Coleman: Consults for Nura Bio, but no support provided for this work. The other authors declare that no competing interests exist.

## Funding

| Funder | Grant reference number | Author |
| --- | --- | --- |
| Biotechnology and Biological Sciences Research Council | BB/S009582/1 | Jonathan Gilley Oscar Jackson Michael P Coleman |
| Wellcome Trust | 220906/Z/20/Z | Jonathan Gilley Oscar Jackson Menelaos Pipis Mary M Reilly Michael P Coleman |
| National Institutes of Neurological Diseases and Stroke and office of Rare Diseases | U54NS065712 | Menelaos Pipis Mary M Reilly |
| National Institute of Neurological Disorders and Stroke | 5R01NS072248-10 | Matt C Danzi Stephan Zuchner |
| National Institute of Neurological Disorders and Stroke | 5R01NS105755-03 | Matt C Danzi Stephan Züchner |
| Medical Research Council | MR/L501529/1 | Ammar Al-Chalabi |
| Medical Research Council | MR/R024804/1 | Ammar Al-Chalabi |
| Economic and Social Research Council | ES/L008238/1 | Ammar Al-Chalabi |
| National Institute of Environmental Health Sciences | K23ES027221 | Stephen A Goutman |
| Motor Neurone Disease Association | | Ammar Al-Chalabi Alfredo Iacoangeli |
| NIHR Biomedical Research Centre, Royal Marsden NHS Foundation Trust/ Institute of Cancer Research | | Ammar Al-Chalabi Alfredo Iacoangeli |
| EU Joint Programme – Neurodegenerative Disease Research | | Ammar Al-Chalabi |
| Robert Packard Center for ALS Research, Johns Hopkins University | | Jonathan Gilley Michael P Coleman |
| Wellcome Trust | 216596/Z/19/Z | John Cooper-Knock |

The funders had no role in study design, data collection and interpretation, or the decision to submit the work for publication.

## Author contributions

Jonathan Gilley, Conceptualization, Data curation, Formal analysis, Investigation, Methodology, Project administration, Supervision, Validation, Visualization, Writing – original draft, Writing – review and editing; Oscar Jackson, Formal analysis, Investigation, Writing – review and editing; Menelaos Pipis, Johnathan Cooper-Knock, Ziv Gan-Or, Formal analysis, Resources, Writing – review and editing; Mehrdad A Estiar, Formal analysis, Resources; Ammar Al-Chalabi, Stephen A Goutman, Leandro Lima, Resources, Writing – review and editing; Matt C Danzi, Formal analysis, Writing – review and editing; Kristel R van Eijk, Alfredo Iacoangeli, Jishu Xu, Formal analysis; Matthew B Harms, Henry Houlden, Julia Kaye, Queen Square Genomics, John Ravits, Guy A Rouleau, Rebecca Schüle, Stephan Züchner, Resources; Mary M Reilly, Conceptualization, Funding acquisition, Resources, Writing – review and editing; Michael P Coleman, Conceptualization, Funding acquisition, Project administration, Supervision, Writing – original draft, Writing – review and editing

**Author ORCIDs**
Jonathan Gilley (iD) http://orcid.org/0000-0002-9510-7956
Oscar Jackson (iD) http://orcid.org/0000-0002-1825-9331

**Ethics**
Human subjects: This is a retrospective study using anonymised data so specific consent was not obtained by the authors, but informed consent and consent to publish was obtained at each site that contributed patient information to this study in accordance with their local Institutional Review Boards (IRBs).

**Decision letter and Author response**
Decision letter https://doi.org/10.7554/eLife.70905.sa1
Author response https://doi.org/10.7554/eLife.70905.sa2

---

## Additional files

**Supplementary files**
• Transparent reporting form

**Data availability**
Genomic data was requested from a variety of previously published datasets from whom interested researchers can request access: Project MinE (https://www.projectmine.com/research/data-sharing/); Answer ALS (https://www.nygenome.org/als-consortium/); GENESIS (https://neuropathycommons.org/genetics/genesis-platform); UCL rare disease (neurology) dataset (available on request from Prof. Henry Houlden); HSP study (available on request from Dr. Rebecca Schüle); Lothian Birth Cohort (https://www.ed.ac.uk/lothian-birth-cohorts/data-access-collaboration). Further information about how to gain access to these datasets and any restrictions on who can gain access to the data is provided on these websites. The specifics of the datasets used are outlined in the Materials and Methods section, and are listed in Tables 1–3 and Figure 8. Source data files of processed numerical data and raw blot images have been provided for Figures 2, 3, 4, 5, 6, 7 and 8, and Figure 2—figure supplement 2, Figure 3—figure supplement 2 and Figure 6—figure supplements 1 and 2.

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
