## [Editor Report]

This manuscript reports the discovery of an enrichment of SARM1 variants with constitutively active NADase activity among patients with ALS and related motor neuron diseases. The genetic evidence is substantiated by functional evidence showing consistent gain of function amongst the disease-associated variants. These findings draw renewed attention to a role for axonal degeneration in some forms of motor neuron disease and provide a rationale for SARM1-directed therapeutic intervention.

---

## [Decision Letter]

**Decision letter after peer review:**

[Editors’ note: the authors submitted for reconsideration following the decision after peer review. What follows is the decision letter after the first round of review.]

Thank you for submitting the paper "Enrichment of SARM1 alleles encoding variants with constitutively hyperactive NADase in patients with ALS and other motor nerve disorders" for consideration at *eLife*. Your submission has been reviewed by 2 peer reviewers and the evaluation has been overseen by a Senior Editor and J. Paul Taylor as the Reviewing Editor. Although the work is of interest, we regret to inform you that the findings at this stage are too preliminary for further consideration at *eLife*.

Specifically, the reviewers and reviewing editor thought that the findings described in this manuscript are potentially very important, underscoring a role for axonal degeneration in some forms of motor neuron disease and providing rationale for SARM1-directed therapeutic intervention. There were some suggestions for strengthening the functional data, but for the most part this aspect of the manuscript was considered a strength. On the other hand, the genetic evidence was considered insufficiently persuasive to support publication at this time. This issue was discussed in some detail and this resulted in some specific suggestions for approaches to strengthening this aspect of the study, including validation in independent cohorts (several large cohorts exist that are likely to be receptive), and functional evaluation of a greater number of variants, including not only those unique to cases, but also those shared by cases and controls. Evidence of gain of function in variants shared by cases and controls (or unique to controls) would undermine the significance of the association.

*Reviewer #1:*

This work initially investigates the possible role of SARM1 in the pathogenesis of ALS by testing for an enrichment of rare alleles within cases and controls from the Project Mine Consortium dataset. Here, they identify an increase in rare variants within the autoinhibitory ARM domain of SARM1. Functional assays were employed to demonstrate that several of the variants display constitutively active properties. Further evaluation of additional databases identified the enriched alleles in ALS and HSP cases. The authors conclude that the alleles represent risk alleles for ALS and other motor nerve disorders.

Although the identification of risk factors for ALS and elucidating their pathogenic mechanism is of high relevance, several of the conclusions are not fully supported by the data analysis and need to be extended.

1. The testing for enrichment of SARM1 alleles within cases vs. controls in the Project Mine dataset was not performed at an unbiased genome-wide level but rather at the candidate gene level. This is reminiscent of candidate gene association testing prior to the advent of GWAS. Unfortunately, >90% of published candidate gene associations were shown to be false-positives once GWAS studies were conducted. As such, caution must be taken in interpreting these results. If indeed a genome-wide multiple test correction were applied to the association testing, the results as presented would not be statistically significant. Further, odds ratio and 95% confidence intervals resulting from the data analysis should be presented.

2. The association testing limits itself to only testing variants within the Project Mine dataset. This dataset is limited in the number of controls and may be underpowered to achieve genome-wide significance. Data from the additional sets (gnomAD, AnswerALS, Genesis) described within the manuscript could be incorporated to improve the burden testing. Please note that caution should be taken in combining aggregate data sets to ensure that functional equivalent pipelines and variant calling parameters were used. This is especially the case for rare variants and difficult to call variants (i.e. indels). It is ideal that identified rare variants, especially indels, are validated using Sanger sequencing or other method.

3. The study does not appear to address possible effects of population stratification or other quality control parameters (e.g. sample call rates, variant call rates, etc.) within any of the data sets used for the burden testing.

4. Several aspects of the functional assay were concerning in supporting the conclusions of the manuscript. Most of the functional experiments were performed in HEK293 cells. For such experiments, a neuronal cell line would be more appropriate. More importantly though, the near total lack of expression of SARM1 variant forms displaying the proposed gain of function activity is most concerning. The reason for the decreased expression was not rigorously addressed. For instance, was the lower expression due to lower RNA expression, increased protein degradation or even possibly due to protein aggregation that may not be detected under the immunoblotting conditions. Alternatively, given that a C-terminal Flag epitope tag was used, transcription or translation could have been prematurely terminated. Establishing the cause of the lower level of expression may shed light on understanding the mechanism of these variants.

5. Similarly, the suppression of expression of ZsGreen (and subsequently DsRed) which is expressed through an IRES element within the SARM1 expression construct, was not appropriately addressed. The rational that this is due to the gain of function properties of the SARM1 variants is not supported by the data presented.

6. Several figures lacked a vector only control and none of the figures displayed the position of molecular weight markers. Further, the level of exogenous vs. endogenous SARM1 expression was not compared throughout the study. As such, it is difficult to assess the possible contribution of the endogenous protein towards the results observed. This information may be valuable is assessing the mechanism of the variant forms.

7. The manuscript briefly mentions that SARM1 forms an octamer complex (Nature 588:658-663). As such, the possibility exists that the SARM1 variants are also contributing their effects through a dominant negative mechanism. It is not known if the SARM1 variants can bind to the WT SARM1.

8. The study microinjects expression constructs into SARM1-/- primary neurons to investigate the response when expressing WT or variant SARM1. Initial experiments demonstrated that injection with WT SARM1 is enough to restore injury-induced neurite degeneration to SARM1-/- neurons. However, the variant SARM1 was not individually expressed into the SARM1-/- but only co-expressed with WT SARM1. Evaluating the SARM1 variant's ability to restore injury-induced neurite degeneration to SARM1-/- neurons would provide additional insight into its mechanism.

9. The manuscript states that injection of the SARM1 constructs was hampered by the limited ability to assess neuron survival. An alternative approach commonly used to study the effect of mutant ALS genes is by longitudinal imaging of primary neurons transfected with expression constructs for ALS associated genes (J. Neurosci. 30:639-649 (2010)).

10. The manuscript describes association testing of only the initial 8 rare variants observed in the Project Mine dataset within the Answer ALS and Genesis datasets. The testing includes the original Project Mine dataset and therefore do not represent an independent replication set. Further, burden testing of only the initial 8 rare variants is biased for the original hypothesis that there is an enrichment in rare variants in the ARM domain. Such tests should include all rare variants identified in the ARM domain observed within cases and controls, not just the initially observed variants.

If indeed there is a true association of rare variants in the ARM domain of SARM1 and ALS, the manuscript will be of high impact. In addition to addressing the concerns above, several additional experiments are suggested to strengthen the manuscript:

11. As noted, GWAS have detected an association between a variant within the SARM1 region (rs35714695; Nature Gen. 48: 1043-1048) and ALS. Given the enrichment of the delta226-232 (rs782325355), it would of interest to test whether these 2 variants are in linkage disequilibrium with each other.

12. Allele frequency data from gnomAD was shown for the alleles detected in the Project Mine dataset. Although not optimal, incorporating this large dataset into the burden testing may help validate the enrichment. However, a quick evaluation of the gnomAD database shows that there are many rare alleles present within the ARM domain (amino acids 61-397) of SARM1 with a combined allele frequency greater than that observed in the Project Mine dataset (https://gnomad.broadinstitute.org/gene/ENSG00000004139?dataset=gnomad_r3 ).

13. The optimal analysis for enrichment would be to obtain raw sequencing data from large number of cases and controls (Project Mine, AnswerALS, Genesis, and additional datasets from dbGaP and EGA). The raw data can then be harmonized together and allow for proper QC. However, this deep of an analysis is quite intensive.

14. Table 1 is confusing in several aspects. The table may be clearer by presenting the chromosome position, major/alternative alleles, dbSNP ID, protein effect and allele frequencies within the cases and controls (as opposed to allele count).

15. In the functional assays, incorporation of a constitutively active form of SARM1, previously created by mutating the ARM domain (Nature 588:658-663), would be useful to demonstrate similar gain of function activities.

*Reviewer #2:*

The authors screened patient data for mutations associated with ALS and found the auto-inhibitory ARM domain of Sarm1 to be important. This is the domain that keeps Sarm1, an NAD+ hydrolase that drive degeneration, off. Interestingly, many of these mutations were found to increase the ability of Sarm1 to drive NAD+ loss in HEK cells, and efficiently degrade NAD+ when purfied and assayed *in vitro*. This is consistent with the notion that they are GOF. In agreement with this effect being via the ARM domain, many are insensitive to further stimulation by NMN, which is thought to activate Sarm1 through destabilizing ARM-TIR interactions. Overall, this is an rigorous study that makes a compelling case for these ALS-associated mutations being GOF. Now the question is, do these mutations cause ALS?

It is quite puzzling that many of these mutations are tolerated in patients to such an age, when many are more potent than the ARM-deleted (thought to be fully-activated) version of Sarm1. This is likely to change how we think about Sarm1 regulation.

It is important to consider that the amounts of NMN used (10 or 50 uM) are quite high. Cellular levels are thought to be much lower.

Please discuss the caveats of the NMN studies.

Can any prediction be made from the recent structural work on Sarm1 to inform how these variants might alter the ARM domain?

How do patients live so long with these mutations?

Did the authors generate any NADase-dead (E642A) versions of these molecules? If so, did that completely block their effects?

---

## [Author Response]

[Editors’ note: The authors appealed the original decision. What follows is the authors’ response to the first round of review.]

The reviewers and reviewing editor thought that the findings described in this manuscript are potentially very important, underscoring a role for axonal degeneration in some forms of motor neuron disease and providing rationale for SARM1-directed therapeutic intervention. There were some suggestions for strengthening the functional data, but for the most part this aspect of the manuscript was considered a strength. On the other hand, the genetic evidence was considered insufficiently persuasive to support publication at this time. This issue was discussed in some detail and this resulted in some specific suggestions for approaches to strengthening this aspect of the study, including validation in independent cohorts (several large cohorts exist that are likely to be receptive), and functional evaluation of a greater number of variants, including not only those unique to cases, but also those shared by cases and controls. Evidence of gain of function in variants shared by cases and controls (or unique to controls) would undermine the significance of the association.

We have now expanded our analysis to a number of new cohorts that include an additional 4,061 ALS and other motor nerve disorder patients and found additional cases with strong GoF *SARM1* alleles. In contrast, there are still no carriers of these alleles within the new control groups. In total, we now have 11,117 cases and 10,402 confirmed ALS-free controls in our combined datasets and we have matched their ancestry as far as is reasonably possible. Another large exome cohort we explored appears to have poor representation of the ARM domain, as indicated by complete absence of the most common, and probably functionally neutral, ARM domain variant (P332Q), so this dataset was not included. Crucially, the confirmed gain-of-function (GoF) variants remain patient-specific in the expanded cohorts, and if the statistics are recalculated with inclusion of data from the best-matched gnomAD population, this further strengthens the significance, despite the likelihood of there being some ALS cases within the gnomAD ‘non-neuro’ population that would weaken this conclusion.

We acknowledge that a hypothesis-driven approach has some caveats as there is a reliance on the strength of the hypothesis and the functional data supporting it. However, both are very strong in this study. We now address this in Discussion paragraph 3, where we also discuss the essential role of well-founded, function-led studies such as this in closing the heritability gap, and the crucial role of public WGS data in Project MinE for this. These complement unbiased, genome-wide studies because rare, but genuinely pathogenic variants may otherwise be impossible to identify because even the largest datasets lack sufficient power.

As suggested, we have performed functional evaluation of more variants that are present either just in controls (3) or just in patients (2) (Figure 4). None of the control-specific alleles shows strong GoF. We have also addressed all other comments from both reviewers below, including the addition of substantial amounts of new data. To accommodate this new data and provide more clarity with respect to our main hypothesis we have substantially re-structured and re-written parts of the manuscript.

Reviewer #1:This work initially investigates the possible role of SARM1 in the pathogenesis of ALS by testing for an enrichment of rare alleles within cases and controls from the Project Mine Consortium dataset. Here, they identify an increase in rare variants within the autoinhibitory ARM domain of SARM1. Functional assays were employed to demonstrate that several of the variants display constitutively active properties. Further evaluation of additional databases identified the enriched alleles in ALS and HSP cases. The authors conclude that the alleles represent risk alleles for ALS and other motor nerve disorders.Although the identification of risk factors for ALS and elucidating their pathogenic mechanism is of high relevance, several of the conclusions are not fully supported by the data analysis and need to be extended.1. The testing for enrichment of SARM1 alleles within cases vs. controls in the Project Mine dataset was not performed at an unbiased genome-wide level but rather at the candidate gene level. This is reminiscent of candidate gene association testing prior to the advent of GWAS. Unfortunately, >90% of published candidate gene associations were shown to be false-positives once GWAS studies were conducted. As such, caution must be taken in interpreting these results. If indeed a genome-wide multiple test correction were applied to the association testing, the results as presented would not be statistically significant. Further, odds ratio and 95% confidence intervals resulting from the data analysis should be presented.

We appreciate the need for caution and have noted in the text the caveats of using a hypothesis-led approach. However, the hypothesis that SARM1 contributes to ALS is already based in part on genome-wide significant association of a nearby SARM1 intragenic SNP in GWAS studies, and well-founded, hypothesis-led studies have an essential role in closing the heritability gap that we also now discuss (Discussion paragraph 3). This may be the only realistic way to identify true pathogenic roles of many rare variants, when even the largest datasets lack sufficient power to rule them either in or out at genome-wide level, and co-ordinating even larger ones is impractical. Moreover, the genome-wide rare variant burden analysis previously used for loss-of-function variants, for example with KIF5A, NEK1, etc, cannot be done for GoF variants as very few can be predicted from sequence alone. However, despite their rarity, their identification is important, because evidence of harmful SARM1 GoF associated with ALS supports the notion that other causes of SARM1 activation could also be involved, for example downstream of STMN2 depletion or SARM1 activation by environmental toxins such as that we recently reported https://www.biorxiv.org/content/10.1101/2020.09.18.304261v2.

Regarding multiple test correction, the whole genome burden test at gene level is not as demanding as GWAS as it is corrected by the number of genes rather than number of SNPs. While we agree there could still be random associations, the p value of 0.00017 is quite convincing, especially considering that a more common SARM1 intragenic SNP that does show genome-wide significant association with ALS (RS35714695) is located just 8-11kb from most of these variants and the ones we include in this calculation all show confirmed GoF. This contrasts with the complete lack of association in neighbouring variants that we found are not strong GoF and illustrates why inclusion of these in the burden test risks masking a genuine pathogenic role of the rare hyperactivating variants.

A true odds ratio (OR) and 95% confidence interval (CI) cannot be calculated for burden tests, as different variants have different effect sizes and different directions. Calculation of an accurate OR is also problematic since the number of GoF variants among the controls is zero. Adding a continuity correction of 0.5 would give an OR well into double figures but also a huge CI ranging from very low to several hundred, which would be of little use.

2. The association testing limits itself to only testing variants within the Project Mine dataset. This dataset is limited in the number of controls and may be underpowered to achieve genome-wide significance. Data from the additional sets (gnomAD, AnswerALS, Genesis) described within the manuscript could be incorporated to improve the burden testing. Please note that caution should be taken in combining aggregate data sets to ensure that functional equivalent pipelines and variant calling parameters were used. This is especially the case for rare variants and difficult to call variants (i.e. indels). It is ideal that identified rare variants, especially indels, are validated using Sanger sequencing or other method.

Only the domain-by-domain tests described in the first Results section were restricted to the Project MinE DF1 dataset alone, while the tests in the penultimate Results section were performed on aggregated datasets. Crucially, the first tests were only performed as a broad indicator that coding variation in the ARM domain might be relevant in ALS, prior to any functional data being obtained, whereas it was the later tests that were used to specifically assess the strength of our main hypothesis that SARM1 GoF is associated with ALS and other motor nerve disorders. Therefore, to avoid possible confusion, we have omitted the first tests from the revised manuscript (as they are anyhow effectively rendered superfluous by the later tests), and we have also re-structured the first Results section for greater clarity with regard to generation of our main hypothesis.

More specifically, data from Answer ALS and GENESIS were already included in the burden/SKAT-O tests reported on page 13 in the original manuscript. In the revised manuscript we have now also added patients and controls from Project MinE Data Freeze 2, a UCL rare disease dataset and an HSP cohort, along with additional controls from the Lothian Birth Cohort. We are cautious about including gnomAD because, unlike these well-phenotyped cohorts, gnomAD does not exclude individuals who may have ALS. Even the ‘non-neuro’ filter only excludes samples obtained from neurological or psychiatric studies, so with a lifetime risk of ALS of around 1/350 it is almost certain there will be many ALS cases even in this filtered group. We have, however, run a separate comparison that shows significant enrichment of the confirmed SARM1 GoF variants relative to the best matched background population (European non-Finnish, non-neuro) in gnomAD v3.1.1 (reasons for specifically using this release are provided in the Materials and methods in the revised manuscript) consisting of 30,000+ individuals. This actually strengthens the association (*p* < 0.000057). As such, the association testing now described in the penultimate Results section is much more thorough, and data for individual cohorts and aggregate data can now be found in two new tables (Table 3 and Table 4 in the revised manuscript). We agree about the need for caution in combining datasets and have taken steps to address this (see below). Unfortunately, we do not have access to the DNAs to perform Sanger sequencing (now noted in the Materials and methods).

3. The study does not appear to address possible effects of population stratification or other quality control parameters (e.g. sample call rates, variant call rates, etc.) within any of the data sets used for the burden testing.

Testing for such covariates across all cohorts is not feasible because this will require access to individual level data. In addition, we have done all that can reasonably be done to address population stratification by using aggregate data from predominantly European datasets and by comparing study dataset frequencies with the European non-Finnish gnomAD population (Table 4 in the revised manuscript). The frequency of the relatively more common P332Q allele, which has a higher frequency in Europeans than other populations, broadly confirms that each dataset is predominantly European or contains a substantial proportion of individuals of European ancestry, as discussed in the Database resources section of the Materials and methods and the Table 3 legend.

4. Several aspects of the functional assay were concerning in supporting the conclusions of the manuscript. Most of the functional experiments were performed in HEK293 cells. For such experiments, a neuronal cell line would be more appropriate. More importantly though, the near total lack of expression of SARM1 variant forms displaying the proposed gain of function activity is most concerning. The reason for the decreased expression was not rigorously addressed. For instance, was the lower expression due to lower RNA expression, increased protein degradation or even possibly due to protein aggregation that may not be detected under the immunoblotting conditions. Alternatively, given that a C-terminal Flag epitope tag was used, transcription or translation could have been prematurely terminated. Establishing the cause of the lower level of expression may shed light on understanding the mechanism of these variants.

Whilst the initial screening for altered SARM1 function was performed in HEK cells, as a convenient way to determine changes in a human cell line that is highly amenable to transfection, the major functional conclusions were actually obtained on recombinant protein and in primary cultured neurons. Importantly, we specifically chose to purify recombinant protein from HEK cells, rather than bacterial or insect cells, as they should support any mammalian-specific post-translational modification that might be critical for NADase activity.

We have addressed the points raised about the low level of expression of GoF variants with new data in Figure 2—figure supplement 2 and additional discussion within the main text in the context of a recent report (Izadifar et al., 2021, PMID: 34384519) that describes a similar effect with overexpression of WT SARM1. That study established that SARM1 overexpression suppresses *de novo* protein synthesis in an NADase-dependent manner. Our new data, showing that expression of the strong GoF SARM1 variants and co-expressed ZsGreen is increased when NAD^+^ production is boosted by nicotinamide riboside supplementation, is entirely consistent with this model. Importantly, this model is further supported by other observations throughout the paper, which we now point out. We also rule out an involvement of truncated form of the variants in question by immunoblotting with antibodies against both the N- and C-terminus of SARM1-Flag. We consider that a low level of expression of full-length variants in HEK cells actually reinforces their strong GoF credentials – essentially much greater NAD consumption is achieved by much less protein. We agree that this a potentially important and relevant observation and this is the reason we highlighted it in the original manuscript. Surprisingly, other manuscripts reporting SARM1 GoF variants / mutants do not appear to report levels of protein expression in (untreated) cells, but we expect they would see a similar effect.

5. Similarly, the suppression of expression of ZsGreen (and subsequently DsRed) which is expressed through an IRES element within the SARM1 expression construct, was not appropriately addressed. The rational that this is due to the gain of function properties of the SARM1 variants is not supported by the data presented.

The reasons for suppression of ZsGreen expression appear to be the same as for the strong GoF variants themselves and are addressed in the responses to comment 4.

6. Several figures lacked a vector only control and none of the figures displayed the position of molecular weight markers.

All experiments were performed with vector only controls (empty vector). In the HEK cell assays all data was normalised to data from cells transfected with empty vector alone (set at 1, and now shown as a dotted line on graphs). In the NADase assays of recombinant proteins, immunoprecipitations from cells transfected with empty vector were performed in parallel to assess non-specific pull-down of contaminating NAD-consuming activities. A proportionately insignificant NAD^+^-consuming activity was seen for these immunoprecipitations. These results were already noted in the relevant section of the Materials and methods in original submission. However, it is not possible to convey this activity in direct relation to that of the SARM1 proteins on graphs reporting mol/min/mol enzyme rate as there is no measurable SARM1 in these controls immunoprecipitations. For the SCG injection experiments, the failure to show (or even mention) empty vector control data was an oversight. These have now been included as new supplementary data (Figure 6—figure supplement 2).

We have also now added the positions of molecular weight markers for all blots in all figures (relative migration of protein bands and molecular weight markers can also be seen in the source data files for each blot that were included with the original submission).

Further, the level of exogenous vs. endogenous SARM1 expression was not compared throughout the study. As such, it is difficult to assess the possible contribution of the endogenous protein towards the results observed. This information may be valuable is assessing the mechanism of the variant forms.

Endogenous SARM1 expression in HEK cells is barely detectable under the conditions we use that can readily detect exogenous WT SARM1 and variants. We have now included a representative blot in Figure 2—figure supplement 1B and discuss what this means for our assays in the main text and relevant figure legends. Essentially endogenous should not make a significant contribution in the HEK cell-based assays or for recombinant protein purified from HEK cells (as a result of co-precipitation in oligomers with the exogenous SARM1).

The situation in injected SCG neurons is less clear, but the amount of expression construct was titrated to minimise pro-death effects of WT SARM1 due to overexpression so it is likely that endogenous SARM1 (which is easily detected by immunostaining using a SARM1 antibody) will be more abundant than the low level of exogenous WT or GoF SARM1 (which cannot be readily detected by immunofluorescence). We have included a comment in the text to that effect, but unfortunately, a direct quantification of endogenous and exogenous SARM1 expression levels in injected SCGs is technically very challenging and would be unreliable.

7. The manuscript briefly mentions that SARM1 forms an octamer complex (Nature 588:658-663). As such, the possibility exists that the SARM1 variants are also contributing their effects through a dominant negative mechanism. It is not known if the SARM1 variants can bind to the WT SARM1.

On this point, we believe the reviewer might have been referring to a possible dominant-positive effect rather than dominant-negative effect (which if it occurred, would be protective not harmful). We previously showed some data supporting this possibility (Figure 3A and 3B in the orginal, now Figure 5A and 5B) but have investigated this in more detail and now provide additional data. This includes co-immunoprecipitation data showing strong GoF SARM1 can oligomerise with WT SARM1 (Figure 5C) and that hetero-oligomers of the two have an intermediate activity (Figure 5D) suggesting that each contributes their individual activity equally to the complex. Importantly, hetero-oligomers of WT and strong GoF SARM1 still have much higher activity than WT SARM1 alone and this explains the "dominant" effect of expression of the strong GoF SARM1 alongside endogenous WT SARM1 in SCG neurons.

8. The study microinjects expression constructs into SARM1-/- primary neurons to investigate the response when expressing WT or variant SARM1. Initial experiments demonstrated that injection with WT SARM1 is enough to restore injury-induced neurite degeneration to SARM1-/- neurons. However, the variant SARM1 was not individually expressed into the SARM1-/- but only co-expressed with WT SARM1. Evaluating the SARM1 variant's ability to restore injury-induced neurite degeneration to SARM1-/- neurons would provide additional insight into its mechanism.

SCG neuron experiments reported in the main figures are restricted to wild-type cultures so that the GoF variant is expressed on a background of endogenous SARM1 to mirror, as best we can, the situation in heterozygous individuals who will express both wild-type and variant SARM1 together. This is outlined in the main text. One of the figure supplements describes the ability of WT SARM1, but not enzymatically-inactive E642A SARM1, to restore axon degeneration in *Sarm1*^-/-^ neurons, but this was simply to show that the amount of expression vector we inject provides sufficient expression of WT SARM1 to be functionally relevant. A similar assessment of strong GoF in this context is more difficult given the associated strong pro-death effect on cell bodies and, whilst interesting, the results would not impact the overall conclusions made in the paper as we already demonstrate clear pro-death/degeneration effects of a strong GoF variant in our current assays.

9. The manuscript states that injection of the SARM1 constructs was hampered by the limited ability to assess neuron survival. An alternative approach commonly used to study the effect of mutant ALS genes is by longitudinal imaging of primary neurons transfected with expression constructs for ALS associated genes (J. Neurosci. 30:639-649 (2010)).

We actually consider our injection assay to be very comparable to the assays described in the citation provided and anticipate that the same problems relating to visualisation of cells injected with the GoF construct (which hampers assessing the effects on neuron/neurite survival rather than injection itself) would also apply to transfected primary neurons (as in transfected HEKs). Furthermore, we feel that injection allows for much greater control over several key variables than transfection (control over quantity of construct introduced and an ability to target neurons in a defined location, among others). We alluded to these benefits as being one reason for favouring this assay but have made this clearer in the revised manuscript.

10. The manuscript describes association testing of only the initial 8 rare variants observed in the Project Mine dataset within the Answer ALS and Genesis datasets. The testing includes the original Project Mine dataset and therefore do not represent an independent replication set. Further, burden testing of only the initial 8 rare variants is biased for the original hypothesis that there is an enrichment in rare variants in the ARM domain. Such tests should include all rare variants identified in the ARM domain observed within cases and controls, not just the initially observed variants.

Grouping all rare ARM domain variants together in the burden testing across multiple cohorts, makes the assumption that they are all functionally equivalent, but this is clearly not the case. We have grouped variants together based on strong functional data and we see a strong association with disease for the strong GoF variant group and a complete lack of association for the larger group of other variants that have properties more like WT SARM1. Grouping all variants together therefore would raise the real risk of the non-GoF variants masking a genuine pathogenic role of the smaller number of rare hyperactivating variants.

If indeed there is a true association of rare variants in the ARM domain of SARM1 and ALS, the manuscript will be of high impact. In addition to addressing the concerns above, several additional experiments are suggested to strengthen the manuscript:11. As noted, GWAS have detected an association between a variant within the SARM1 region (rs35714695; Nature Gen. 48: 1043-1048) and ALS. Given the enrichment of the delta226-232 (rs782325355), it would of interest to test whether these 2 variants are in linkage disequilibrium with each other.

This is a very interesting suggestion, but even the most common GoF variant is too rare to give any meaningful indication of linkage disequilibrium (LD) with the GWAS SNP (D’=1; r^2^ = 2.147845e-05). Thus, as we now outline in Discussion paragraph 3, the GWAS and the association with rare GoF variants are independent pieces of data both supporting the involvement of SARM1 in ALS, but each with its own different caveats. Importantly, the reviewer’s question also prompted us to note the close physical proximity of the GWAS SNP and all of the strong GoF SNPs we studied (8.3-11.4kb), all within the same LD block and all within the SARM1 gene so this is now also pointed out in the Discussion.

12. Allele frequency data from gnomAD was shown for the alleles detected in the Project Mine dataset. Although not optimal, incorporating this large dataset into the burden testing may help validate the enrichment. However, a quick evaluation of the gnomAD database shows that there are many rare alleles present within the ARM domain (amino acids 61-397) of SARM1 with a combined allele frequency greater than that observed in the Project Mine dataset (https://gnomad.broadinstitute.org/gene/ENSG00000004139?dataset=gnomad_r3 ).

Thank you for this suggestion. We have considered the burden analysis for known GoF variants both with and without the inclusion of the best matched gnomAD population on page 17, and despite its caveats (that there are likely to be some gnomAD individuals with ALS or HSP) its inclusion does increase the significance. Regarding the other rare ARM domain variants in gnomAD, whilst there are many and together they have a higher combined allele frequency, the fact remains that they are absent in our cohorts. Testing each of them for GoF is also neither feasible nor particularly useful given their extremely low individual frequencies, and including them in the analysis regardless of whether they cause functional change adds unnecessary noise as reasoned above in point 10. The gnomAD dataset also includes several very different ethnic populations so a correct comparison would need to stratify it accordingly. Thus, we prefer to limit the inclusion of gnomAD data to that reported in the penultimate Results section.

13. The optimal analysis for enrichment would be to obtain raw sequencing data from large number of cases and controls (Project Mine, AnswerALS, Genesis, and additional datasets from dbGaP and EGA). The raw data can then be harmonized together and allow for proper QC. However, this deep of an analysis is quite intensive.

We agree with the reviewer that analysing raw sequencing data for all of these and other cohorts would be very intensive and is thus not really feasible within the timeframe of normal revisions. However, there is strong confidence in the calling of the variants tested in this study based on phred-scaled quality scores (when available). We have included a comment in the Materials and methods in this regard.

14. Table 1 is confusing in several aspects. The table may be clearer by presenting the chromosome position, major/alternative alleles, dbSNP ID, protein effect and allele frequencies within the cases and controls (as opposed to allele count).

We now provide substantially more detail about the variants (both ARM domain and other regions) for both Project MinE (data freeze 1) and Answer ALS in Table 1 and Table 2 in the revised manuscript.

15. In the functional assays, incorporation of a constitutively active form of SARM1, previously created by mutating the ARM domain (Nature 588:658-663), would be useful to demonstrate similar gain of function activities.

Constitutive NADase activities obtained from the recombinant protein assays were already compared to the activity of the well-established SAM-TIR fragment lacking the entire auto-inhibitory ARM domain. This is widely considered in the field to have "maximal" activity and be the most suitable reference for GoF. In the original submission the data was rather hidden away in the main text and based on activity measurements from independent experiments. We have now repeated the experiments in parallel with one of the strong GoF variants identified in this study and present the data in Figure 3B in the revised manuscript. Importantly the reported activities in these assays are per mol rates allowing the most accurate comparison possible. This is not possible using indirect measures from transfected cells.

Reviewer #2:The authors screened patient data for mutations associated with ALS and found the auto-inhibitory ARM domain of Sarm1 to be important. This is the domain that keeps Sarm1, an NAD+ hydrolase that drive degeneration, off. Interestingly, many of these mutations were found to increase the ability of Sarm1 to drive NAD+ loss in HEK cells, and efficiently degrade NAD+ when purfied and assayed *in vitro*. This is consistent with the notion that they are GOF. In agreement with this effect being via the ARM domain, many are insensitive to further stimulation by NMN, which is thought to activate Sarm1 through destabilizing ARM-TIR interactions. Overall, this is an rigorous study that makes a compelling case for these ALS-associated mutations being GOF. Now the question is, do these mutations cause ALS?It is quite puzzling that many of these mutations are tolerated in patients to such an age, when many are more potent than the ARM-deleted (thought to be fully-activated) version of Sarm1. This is likely to change how we think about Sarm1 regulation.

We agree and we have discussed this point in paragraph 6 of the Discussion. We consider it very likely that this inconsistency points to compensatory changes during development in the carriers in response to chronic expression of strong NADase activity. At this point we cannot provide any detail about what these changes might be.

It is important to consider that the amounts of NMN used (10 or 50 uM) are quite high. Cellular levels are thought to be much lower.

We acknowledge that our assays of recombinant proteins will not reflect physiological conditions but we have now provided explanations in the relevant parts of the main text as to why we chose these concentrations and how they might relate to physiological levels.

Can any prediction be made from the recent structural work on Sarm1 to inform how these variants might alter the ARM domain?

This is a very interesting suggestion. We have touched on the relationship between the GoF variants and existing structural knowledge in paragraph 5 of the Discussion. However, in our experience this is not a substitute for empirical testing although sometimes it is useful as a guide.

How do patients live so long with these mutations?

As discussed above, this is one of the key questions arising from this work and we allude to it several times in the manuscript. However, as yet, we cannot provide an answer. This will be the focus of follow-up studiesbut we consider it out of the scope of this study.

Did the authors generate any NADase-dead (E642A) versions of these molecules? If so, did that completely block their effects?

We have made a double mutant combining E642A with one of the strong GoF mutations and we describe its effects in some of our key assays in new Figure 7. Most importantly, we show that this blocks the pro-death effects in SCG neurons.